# Paternal restraint stress affects offspring metabolism via ATF-2 dependent mechanisms in *Drosophila melanogaster* germ cells

Ki-Hyeon Seong [1,2,9 ✉], Nhung Hong Ly[1], Yuki Katou[3], Naoko Yokota[4], Ryuichiro Nakato[4], Shinnosuke Murakami [5,6], Akiyoshi Hirayama[5,6], Shinji Fukuda [2,5,6,7], Siu Kang[8], Tomoyoshi Soga[5,6], Katsuhiko Shirahige[3,9] & Shunsuke Ishii [1,9 ✉]

Paternal environmental factors can epigenetically influence gene expressions in offspring. We demonstrate that restraint stress, an experimental model for strong psychological stress, to fathers affects the epigenome, transcriptome, and metabolome of offspring in a MEKK1-dATF2 pathway-dependent manner in *Drosophila melanogaster*. Genes involved in amino acid metabolism are upregulated by paternal restraint stress, while genes involved in glycolysis and the tricarboxylic acid (TCA) cycle are downregulated. The effects of paternal restraint stress are also confirmed by metabolome analysis. dATF-2 is highly expressed in testicular germ cells, and restraint stress also induces p38 activation in the testes. Restraint stress induces Unpaired 3 (Upd3), a *Drosophila* homolog of Interleukin 6 (IL-6). Moreover, paternal overexpression of *upd3* in somatic cells disrupts heterochromatin in offspring but not in offspring from *dATF-2* mutant fathers. These results indicate that paternal restraint stress affects metabolism in offspring via inheritance of dATF-2-dependent epigenetic changes.

[1] RIKEN Cluster for Pioneering Research, Tsukuba, Ibaraki, Japan. [2] PRESTO, Japan Science and Technology Agency, Kawaguchi, Japan. [3] Laboratory of Genome Structure and Function, Institute for Quantitative Biosciences, University of Tokyo, Tokyo, Japan. [4] Laboratory of Computational Genomics, Institute for Quantitative Biosciences, University of Tokyo, Tokyo, Japan. [5] Institute for Advanced Biosciences, Keio University, Tsuruoka, Yamagata, Japan. [6] Systems Biology Program, Graduate School of Media and Governance, Keio University, Fujisawa, Kanagawa, Japan. [7] Transborder Medical Research Center, University of Tsukuba, Tsukuba, Ibaraki, Japan. [8] Department of Bio-Systems Engineering, Graduate School of Science and Engineering, Yamagata University, Yonezawa, Yamagata, Japan. [9] These authors jointly supervised this work: Katsuhiko Shirahige, Shunsuke Ishii. ✉email: ki-hyeon.seong@riken.jp; sishii@riken.jp

Recently accumulated evidence in model organisms indicates that the epigenetic status can affect the next generations apart from DNA sequences in response to several metabolic and environmental cues of their parents[1–3]. High-fat and low-protein diet in rodents alter gene expression and cause modest change of DNA cytosine methylation in offspring[4,5], and sugar diets alter the metabolic and histone modification status of offspring in invertebrates[6,7]. Environmental stresses, such as thermal and osmotic stress, induce epigenetic change of offspring in flies and nematodes[8,9]. Recent studies have suggested that the epigenetic inheritance mediated by paternal stresses might occur through the sperm, and that DNA methylation, histone modifications, and noncoding RNAs in sperm are implicated in the paternal transmission of epigenetic information[10,11]. Some mammalian studies indicate the effects of paternal exposure of environmental stimuli on DNA methylation status of sperm[12,13]. Recent studies also indicate the effect of changes in microRNA and tRNA fragment content in sperm by stresses and diet changes[14–16]. Although a small fraction of the genome escapes the histone/protamine replacement during spermatogenesis, the change of histone modification in testicular germ cells induce by low-protein diet is maintained in mouse sperm[17]. In *Caenorhabditis elegans*, temperature-induced epigenetic changes in heterochromatic histone H3K9 trimethylation (H3K9me3) level occurs primarily in both oocyte and sperm, and is transmitted to multiple generations[9]. These findings show that remaining histones also have the potential to transmit epigenetic information to the next generation. However, the exact genetic and mechanistic basis of environmental stress-induced epigenetic inheritance remains unclear.

Studies using *Drosophila* model have contributed greatly understanding of the epigenetic inheritance induced by parental metabolic and environmental stresses[6,8,18–20]. The DNA cytosine methylation is detected but at a quite low level[21], and the mechanistic contribution of DNA methylation to the epigenetic inheritance has not been observed in the *Drosophila* system. On the other hand, the implications of modified histones and small RNAs for epigenetic inheritance have been demonstrated in some reports[6,8,22,23]. However, it remains still unclear how metabolic and environmental stress can transmit epigenetically to offsprings in *Drosophila*.

In recent years, new evidence has demonstrated epigenetic inheritance of acquired traits caused by parental psychological stress. The exposure to traumatic stress in early postnatal life in mice alters synaptic plasticity in the brain and induces depressive-like behaviors in their offspring[24]. F1 and F2 offspring of F0 male mice conditioned with the odor acetophenone, which activates the odorant receptor *Olfr151*, exhibit increased behavioral sensitivity to acetophenone[25]. Sperm DNA from conditioned F0 males and F1 offspring show CpG hypomethylation in the *Olfr151* gene, suggesting that paternal traumatic exposure is inherited via changes in DNA methylation of sperm DNA. In addition, early life stress of F0 male mice induced by unpredictable maternal separation and maternal stress cause depressive-like behaviors and altered microRNA expression in the sperm of F0 and F1 offspring[26]. Injection of altered microRNAs from the sperm of F0 mice into fertilized wild-type oocytes leads to similar behavioral and metabolic changes in F1 and F2 mice. Furthermore, paternal restraint stress can enhance liver gluconeogenesis in mouse offspring by increasing the level of phosphoenolpyruvate carboxykinase (PEPCK), which is associated with changes in DNA methylation of specific microRNAs in sperm to regulate PEPCK translation[27]. Together, these findings suggest that paternal psychological stress affects traits and gene expression patterns in offspring via inheritance of epigenetic change, but the mechanism remains elusive.

Transcription factor activating transcription factor 2 (ATF2), a member of the ATF/CREB (cAMP responsive element binding) superfamily, binds to the CRE (cAMP response element)[28–31]. The subfamily of ATF2 proteins are phosphorylated by stress-activated protein kinase p38 in response to various stresses, including inflammatory cytokines, oxidative stress, and psychological stress[30,31].

Recently, we have reported that *Drosophila* dATF-2 and vertebrate ATF7, an ATF2 subfamily member, contribute to pericentromeric heterochromatin formation. Heat shock or osmotic stress induces phosphorylation of dATF-2 via p38, which causes a release of dATF-2 from chromatin, resulting in a decrease in the level of histone H3K9 dimethylation (H3K9me2) and heterochromatin disruption. Heterochromatin disruption in male germ cells by heat shock is not completely recovered and is instead transmitted to the next generation, suggesting inheritance of the heat shock stress-induced decrease in H3K9me2[8]. Thus, ATF2 subfamily proteins play a key role in the stress-induced heterochromatin disruption as a stress-responsive epigenetic regulator.

Herein, we explore the role of dATF-2 in paternal psychological stress-induced gene expression changes in *Drosophila* offspring. We demonstrate that paternal restraint stress affects the epigenome, transcriptome, and metabolome status of offspring in a dATF-2-dependent manner. Moreover, our results suggest that restraint stress-induced unpaired 3 (Upd3) activates p38 in testes and affects heterochromatin status in offspring.

## Results

**Paternal restraint stress-induced heterochromatin disruption is dATF-2-dependent**. Restraint stress has long been used primarily as the preferred means to study mammalian mental disorders because it can induce strong psychological stress without pain stress[32]. To expose mice to restraint stress, animals are usually restrained in a plastic tube or bag. We used restraint stress in *Drosophila* to test whether father's psychological stress affects offspring traits. To expose adult males to restraint stress, flies were sandwiched by soft sponge plugs for 10 h per day (Fig. 1a and Supplementary Fig. 1a, b). As controls, flies were maintained freely without medium (Supplementary Fig. 1a). Restraint stress exposure for 10 h per day once or twice did not affect lethality, while restraint stress exposure three times slightly (~20%) increased lethality (Supplementary Fig. 1c). Previously, we showed that heat shock stress disrupts heterochromatin, which is transmitted to the next generation[8]. To investigate the inheritance of restraint stress-induced heterochromatin disruption, we examined position effect variegation (PEV) using the $In(1)w^{m4}$ line (referred to hereafter as $w^{m4}$) as in our previous study[8]. The $w^{m4}$ line, established by backcrossing with $w^{1118}$ for six generations, was used in the present study. Since transient restraint stress may induce epigenetic change in specific types of testicular germ cells, such as spermatagonial stem cells and spermatocytes, we carefully controlled experimental conditions, especially the frequency of restraint stress and the timing of hatching. When male flies (F0) were exposed to restraint stress for 10 h on the 1st day only once and then mated on the 4th day, the first brood generated by mating for 3 days exhibited increased eye pigment in male F1 flies following paternal restraint stress (Fig. 1c). However, the following three successive broods did not exhibit increased eye pigment phenotype. Restraint stress exposure twice also yielded similar results (Fig. 1d). Repetitive restraint stress exposure three times resulted in relatively high amount of eye pigment not only in the first brood but also in the second and fourth broods, and third brood shows insignificant but higher eye pigment level (Fig. 1b, e). This restraint stress-induced eye pigment

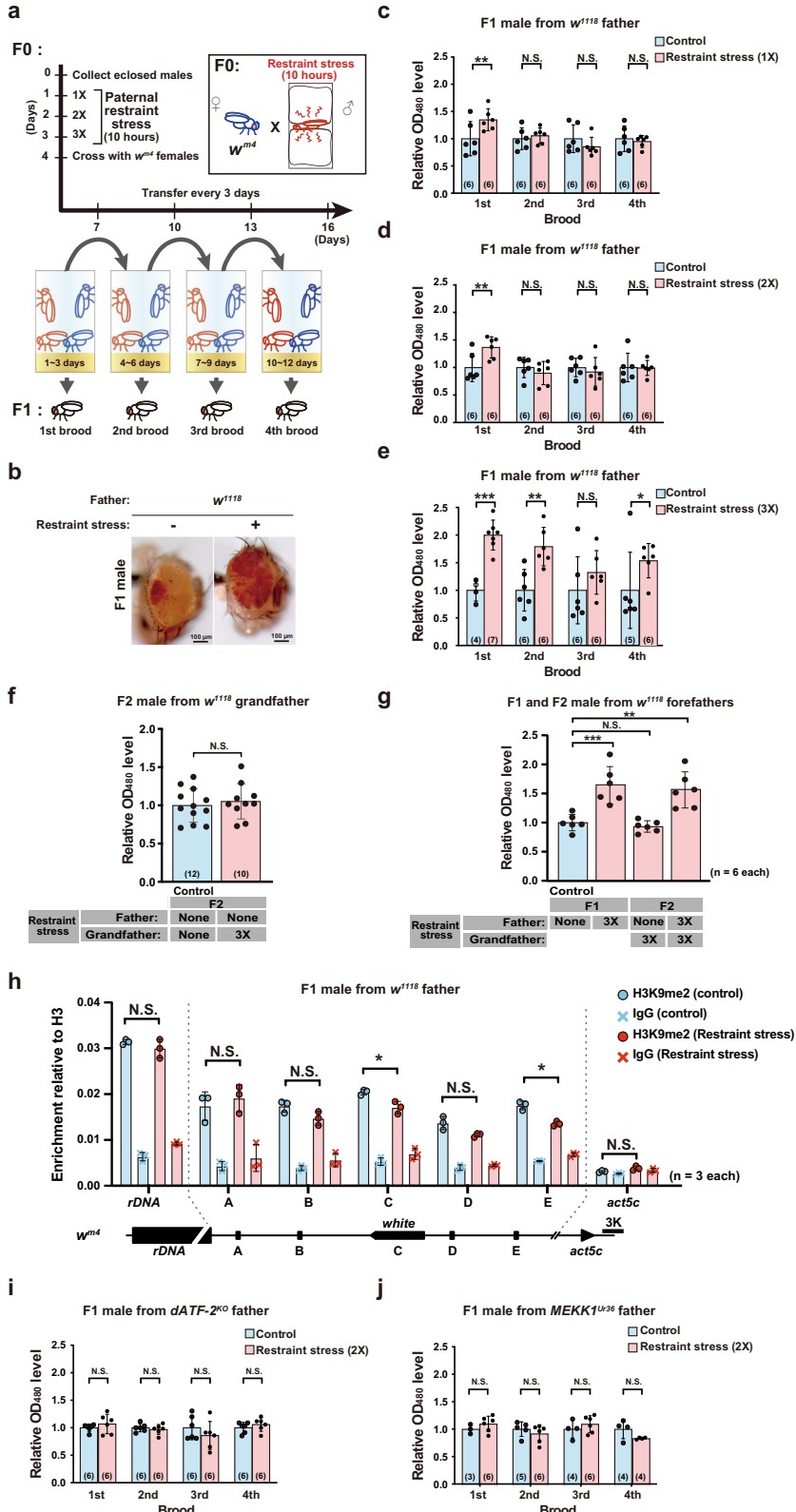

phenotype occurred in both F1 males and F1 females (Supplementary Fig. 1d, e). We explored whether the restraint stress-induced eye pigment phenotype might carry to the F2 generation. Paternal restraint stress did not affect the eye pigment level in the F2 offspring (Fig. 1f), and there was no additive or synergistic effect of restraint stress exposure performed for two consecutive generations (Fig. 1g). These results indicate that the paternal

restraint stress-induced heterochromatin disruption is transmitted to the F1, but not to the F2 offspring. We thought that the paternal restraint stress-induced *w* expression in F1 offsprings might cause epigenetic changes in testicular germ cells of fathers, and the epigenome status might be transmitted to F1 offsprings. We examined the chromatin status of F1 males by chromatin immunoprecipitation (ChIP) assay using primer sets focused on

**Fig. 1 Paternal restraint stress induces heterochromatin disruption in *Drosophila* offspring. a** Schematic diagram of the experimental design. The levels of red-eye pigment were measured in paternal restraint stress-exposed and paternal restraint stress-free (control) $w^{m4}$ F1 progeny. $w^{1118}$ males were exposed to restraint stress for 10 h per day, restraint stress treatment was repeated up to three times, and $w^{1118}$ males were crossed with $w^{m4}$ females. Parents were transferred to new vials every 3 days and removed from vials after 12 days. Red-eye pigment levels in F1 males from each brood were subsequently measured. **b** Eye phenotypes of F1 $w^{m4}$ males derived from $w^{1118}$ fathers with/without restraint stress. Red-eye pigment levels, measured as the absorbance/optical density (OD) at 480 nm, in F1 male progeny from fathers exposed to paternal restraint stress once (**c**), twice (**d**), or three times (**e**). The value of red-eye pigment represents relative to control of F1 male progeny, which derived from restraint stress-free fathers. Averages with standard deviation (s.d.) are shown (\*\*\*$p < 0.001$; \*\*$p < 0.01$; \*$p < 0.05$; N.S., no significant difference, Student's unpaired $t$ test). **f** Red-eye pigment levels of F2 male derived from $w^{1118}$ grandfather with/without restraint stress. **g** Red-eye pigment levels of F1 and F2 males derived from $w^{1118}$ father and grandfather with/without restraint stress. **h** ChIP results performed using anti-H3K9me2, anti-H3, and control IgG. The value of $y$-axis represents the ratio of immunoprecipitated DNA with anti-H3K9me2 or control IgG to that with anti-H3 antibody. A diagram of the rearranged white locus on X chromosome is represented in the bottom panel, and target regions for ChIP assays are indicated in the diagram. Averages with s.d. are shown (\*$p < 0.05$; N.S., no significant difference, multiple $t$-test). Red-eye pigment levels in F1 males derived from $dATF\text{-}2^{KO}$ (**i**) and d $Mekk^{1Ur36}$ (**j**) fathers. Averages with s.d. are shown (N.S., no significant difference, Student's unpaired $t$ test). Number of samples analyzed indicated in parentheses on each graph. See also Supplementary Fig. 1.

regions around the rearranged *white* (*w*) gene in $w^{m4}$, an rDNA coding region which is located in a heterochromatic region, and an actively transcribed euchromatic *Act5C* region as described in a previous report[33] (Fig. 1h). We observed that the histone H3K9me2 level of the *w* gene locus was slightly but significantly reduced in F1 males derived from restraint stress-exposed father, while the level in the rDNA region did not show any change. In the *Act5c* region, the histone H3K9me2 binding was extremely low level and was not changed by paternal restraint stress. These results suggest that restraint stress can induce heterochromatin disruption by decreasing the level of histone H3K9me2 in specific types of testicular germ cells, and the degree of heterochromatin disruption appears to vary depending on the frequency of restraint stress. Thus, these results indicate that heterochromatin disruption by restraint stress is inherited by the next generation.

We previously used the $dATF\text{-}2^{PB}$ mutant, in which dATF-2 mRNA level was severely reduced by insertion of the piggyBac transposon into the upstream of the translation initiation codon of $dATF\text{-}2^8$. However, the $dATF\text{-}2$ protein-coding region was intact in this mutant. Therefore, to examine the effects of paternal restraint stress in a $dATF\text{-}2$ null mutant background, we have newly generated a $dATF\text{-}2$ knockout ($dATF\text{-}2^{KO}$) fly line by the ends-out gene targeting method (Supplementary Fig. 2)[34]. We confirmed that the $dATF\text{-}2^{KO}$ flies also showed the same eye phenotype and increased *w* gene expression as $dATF\text{-}2^{PB}$ in $w^{m4}$ background (Supplementary Fig. 2f). The $dATF\text{-}2^{KO}$ homozygous males were exposed to restraint stress twice, and then crossed with $w^{m4}$ females. We observed the F1 offspring from the $dATF\text{-}2^{KO}$ mutants exhibited increased eye pigment phenotype, and paternal restraint stress did not further affect the level of eye pigment (Fig. 1i and Supplementary Fig. 1f). We have performed the same experiment using *Mekk1*-deficient homozygous fathers and observed that paternal restraint stress did not increase eye pigment in their offspring (Fig. 1j and Supplementary Fig. 1g). As dATF-2 is phosphorylated by p38, which is activated by upstream kinase MEKK1 in response to various stresses, these results suggest that inheritance of restraint stress-induced heterochromatin disruption depends on dATF-2 phosphorylation by p38.

**Paternal restraint stress-induced gene expression change is a dATF-2-dependent.** We next examined the effects of paternal restraint stress on gene expression patterns in offspring. To examine this under normal heterochromatin status, we used $w^{1118}$ wild-type flies, not $w^{m4}$ flies. Wild-type male flies were exposed to restraint stress three times and mated with wild-type females (Fig. 2a). RNA was prepared separately from the heads and other parts of offspring, and used for RNA-seq analysis. In head samples, differentially expressed genes with a false discovery

rate (FDR) $q$-value $< 0.05$ and fold change $> 1.4$ following paternal restraint stress were not observed (Supplementary Table 1). However, in samples from other parts, 279 and 1232 genes were up- and downregulated by paternal restraint stress, respectively (Fig. 2b). When male $dATF\text{-}2$ mutant flies were used as fathers, paternal restraint stress upregulated only three genes and downregulated only one gene in samples from other parts (Fig. 2b). These results indicate that dATF-2 is required for paternal restraint stress-induced gene expression changes in offspring.

Genes significantly upregulated by paternal restraint stress were related to eight biological processes, all of which are involved in metabolism, including "glycine, serine, and threonine metabolism" and "one-carbon folate pool" (Fig. 2c). These genes are involved in one-carbon metabolism, which plays an important role in multiple outputs including the biosynthesis of amino acids, nucleotides, glutathione, and taurine (Supplementary Fig. 3a)[35]. Conversely, paternal restraint stress decreased the expression of genes involved in glycolysis/gluconeogenesis and those involved in the tricarboxylic acid (TCA) cycle (Fig. 2c and Supplementary Fig. 3b). Genes involved in approximately half of the reaction steps in glycolysis and in the TCA cycle were downregulated by paternal restraint stress, including rate-limiting enzymes such as hexokinase and pyruvate kinase in glycolysis, and isocitrate dehydrogenase in the TCA cycle (Supplementary Fig. 3b). These results indicate that paternal restraint stress may greatly influence energy metabolism in offspring.

**Effect of paternal restraint stress on metabolites in offspring.** To investigate the relationship between gene expression changes and metabolism, metabolome analysis of whole-body fly samples was performed by capillary electrophoresis time-of-flight mass spectrometry (CE-TOFMS) and liquid chromatography–tandem mass spectrometry (Fig. 3a), and 237 metabolites were identified. To visualize the effects of paternal restraint stress on metabolites in offspring, we performed principal component analysis (PCA) (Fig. 3b). Score plots of metabolome data for offspring from restraint stress-exposed wild-type fathers were clearly different from those of wild-type control flies. By contrast, PCA score plots of metabolome data for offspring from restraint stress-exposed $dATF\text{-}2$ mutant fathers were similar to those of control $dATF\text{-}2$ mutant fathers, indicating that the effect of paternal restraint stress on metabolites in offspring was lost in $dATF\text{-}2$ mutant fathers. Furthermore, PCA score plots of metabolome data obtained using $dATF\text{-}2$ mutant fathers strongly overlapped with those obtained from restraint stress-exposed wild-type fathers. Thus, the effects of restraint stress treatment and $dATF\text{-}2$ mutation in fathers on metabolites in offspring were similar. Since

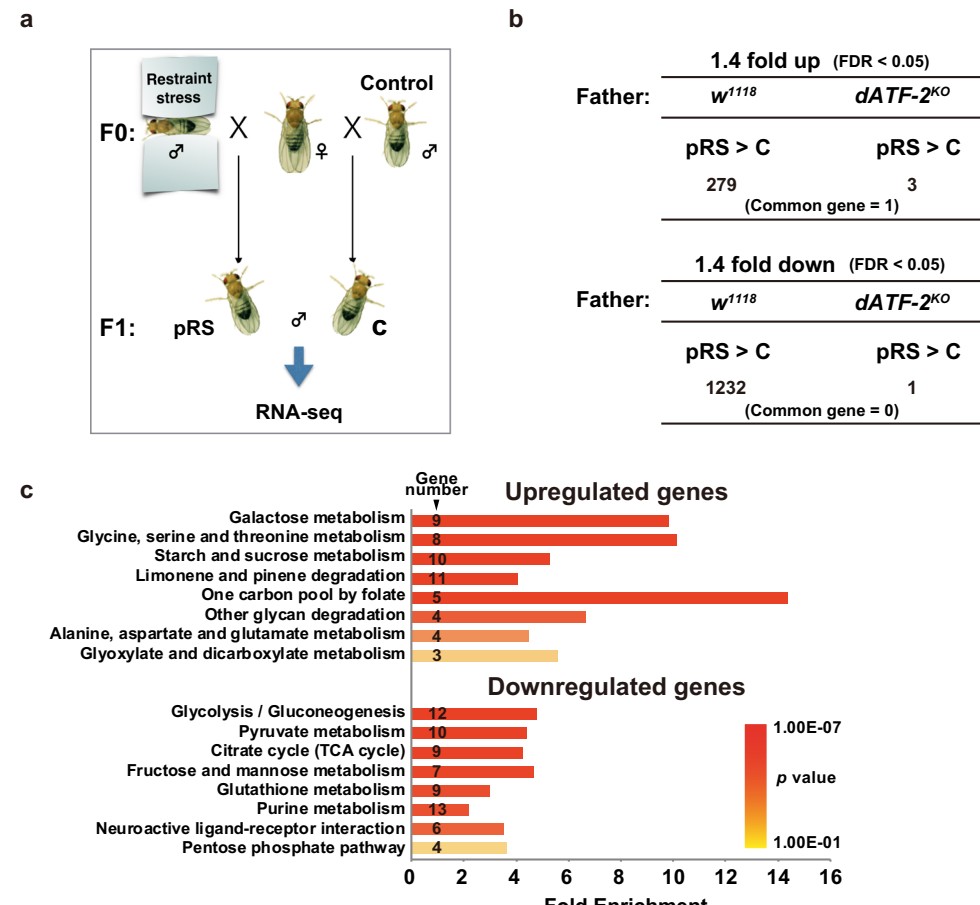

**Fig. 2 Paternal restraint stress alters the transcriptome of *Drosophila* offspring. a** Schematic diagram of RNA sequencing (RNA-seq) sampling. **b** Summary of RNA-seq analysis of different tissues of male F1 flies. Restraint stress exposure of *w1118* fathers altered the expression levels of a group of genes in F1 adult males. However, such effects were not observed when *dATF-2KO* fathers were used. The upper table shows the number of genes upregulated (≥1.4-fold, FDR < 0.05) in offspring by paternal restraint stress treatment of *w1118* and *dATF-2KO* fathers. The lower table shows the number of genes downregulated (≥1.4-fold, FDR < 0.05) by the same treatment. **c** The results of pathway enrichment analysis of up- and downregulated genes in F1 males subjected to paternal restraint stress treatment. See also Supplementary Fig. 3.

stress induces a release of dATF-2 from heterochromatin and target genes via p38 phosphorylation[8], inferring a similar effect of stress and *dATF-2* mutation—both of which cause a loss of dATF-2 on the target genes—is reasonable.

Metabolites were divided into three clusters based on the effects of paternal restraint stress and *dATF-2* mutation on their abundance (Fig. 3c). The levels of metabolites in cluster C1 were decreased by either paternal restraint stress or *dATF-2* mutation, while those in cluster C2 were increased by paternal restraint stress only. The levels of metabolites in cluster C3 were increased by *dATF-2* mutation only. Compared with metabolites in clusters C2 and C3, a decrease in the levels of metabolites in cluster C1 was more significant. Specifically, metabolites linked to the TCA cycle, gluconeogenesis, glycolysis, and the mitochondrial electron transport chain were enriched (Fig. 3d). Typical examples include 2,3-DPG, which is involved in glycolysis; FAD and acetyl CoA, which are linked to the TCA cycle; and ATP and NAD+, which are associated with glycolysis, the TCA cycle, and the mitochondrial electron transport chain (Supplementary Fig. 4). The levels of these metabolites in offspring were decreased by paternal restraint stress and also *dATF-2* mutation in fathers, but there were no additional effects of combined paternal restraint stress treatment and *dATF-2* mutation. Together, the transcriptome and metabolome analyses indicate a strong correlation between paternal restraint stress-induced changes in gene expression and

metabolite content related to energy metabolism, and that paternal restraint stress affected metabolism in offspring in a dATF-2-dependent manner.

**Effect of paternal restraint stress on the phenotypes of offspring.** To investigate the effects of paternal restraint stress on the phenotypes of offspring, we analyzed sensitivity to the drug rotenone, an inhibitor of mitochondrial electron transport chain complex I (Fig. 4a). Rotenone induces dose-dependent ATP depletion and oxidative damage[36]. In *Drosophila*, locomotor impairment and dopaminergic neuron degeneration are induced by rotenone[37]. In the present study, offspring from restraint stress-exposed wild-type fathers were more sensitive to rotenone and died earlier than offspring from control fathers (Fig. 4b). However, this difference in rotenone sensitivity was not observed among offspring from *dATF-2* mutant flies (Fig. 4c). These results suggest that paternal restraint stress decreases ATP levels in offspring, which increases their sensitivity to rotenone.

**Activation of p38 by restraint stress in testicular germ cells.** The results of quantitative real-time PCR (qRT-PCR) (quantitative reverse transcription-polymerase chain reaction) using various fly tissues indicated that dATF-2 mRNA is ubiquitously expressed in multiple tissues and highly expressed especially in the testis (Fig. 5a). In situ hybridization (ISH) showed that

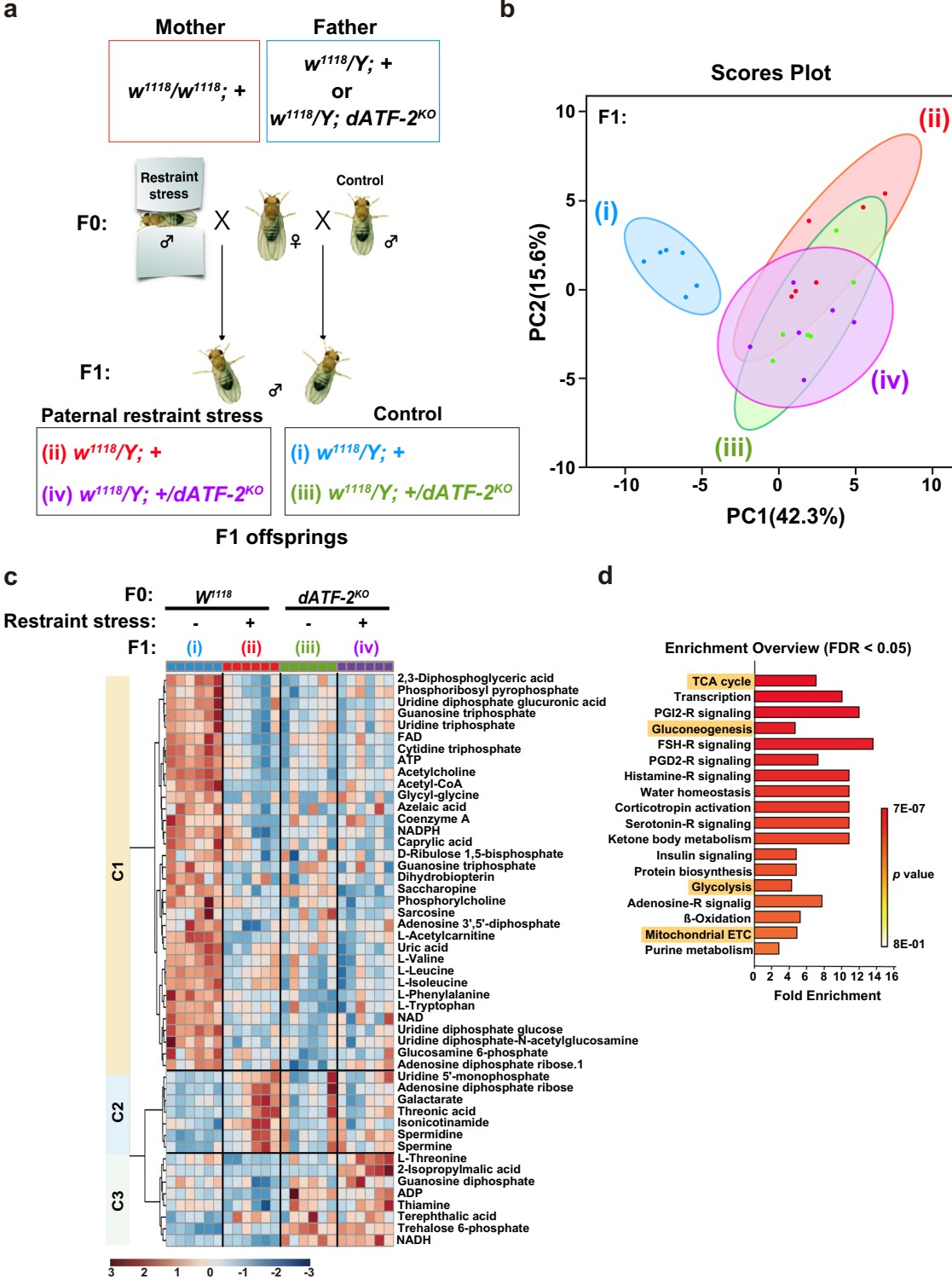

**Fig. 3 Paternal restraint stress affects the metabolome of *Drosophila* offspring. a** Schematic diagram of the cross for metabolome analysis. Each genotype of offspring is represented by the same roman numerals and colors in Fig. 5. **b** Two-dimensional score plot of principal component analysis. Blue and red represent F1 males from *w1118* fathers exposed to paternal restraint stress and controls, respectively; green and purple represent F1 males from paternal restraint stress-exposed or control *dATF-2KO* fathers, respectively. **c** Hierarchical clustering of metabolites affected (both up- and downregulated) by paternal restraint stress. A total of 49 metabolites were significantly affected by paternal restraint stress treatment of *w1118* and *dATF-2KO* fathers. Red and blue indicate high- and low-intensity levels, respectively. **d** Enrichment overview of metabolites significantly affected by paternal restraint stress. Pathways highlighted in yellow are related to glucose energy metabolism. See also Supplementary Fig. 4.

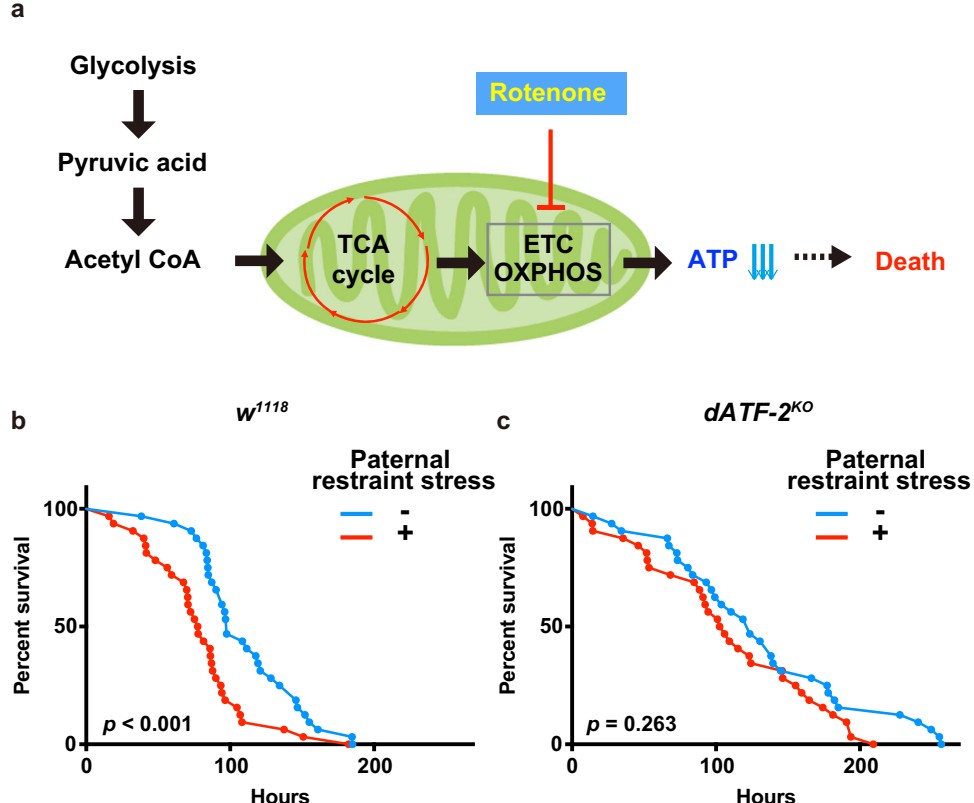

**Fig. 4 Paternal restraint stress enhances rotenone sensitivity. a** Schematic diagram of mitochondrial energy transduction. Rotenone, an inhibitor of the mitochondrial electron transport chain (ETC), reduces ATP levels and triggers fly lethality. Survival curves following rotenone treatment (5 mM) for adult males derived from $w^{1118}$ (**b**) ($n = 32$ each; $p < 0.001$, log-rank [Mantel–Cox] test) and $dATF\text{-}2^{KO}$ (**c**) ($n = 32$ each; $p = 0.263$, log-rank [Mantel–Cox] test) fathers with/without paternal restraint stress.

dATF-2 mRNA is expressed in spermatogonia and spermatocytes (Fig. 5b). We also analyzed level of p38 phosphorylation by western blotting at four time points (1, 5, and 10 h after start of restraint stress and 12 h after end of restraint stress) (Fig. 5c and Supplementary Fig. 5a). The level of phosphorylated p38 (P-p38) in whole-body samples increased during restraint stress exposure (Fig. 5d and Supplementary Fig. 5b). On the other hand, restraint stress-dependent phosphorylation of p38 in the testes was increased after the 10 h restraint stress treatment, but it was not observed after 1 and 5 h restraint stress exposure (Fig. 5e and Supplementary Fig. 5c). At 12 h after the end of restraint stress, no significant increase of P-p38 was observed in both whole body and testes samples, indicating that P-p38 is not maintained for a long period (Fig. 5d, e and Supplementary Fig. 5b, c). We also observed that an increase of P-p38 was repeatedly detected in the testis samples prepared immediately after the second and the third times of 10 h restraint stress treatment (Supplementary Fig. 5d). Immunostaining of P-p38 in the testes revealed that restraint stress increased P-p38 signals in spermatagonial stem cells and spermatocytes (Fig. 5f, g and Supplementary Fig. 6). These results suggest that the activation of p38 in testicular germ cells is a relatively late event, and transient.

**Epigenetic effects of paternal restraint stress-induced Upd-3.** It is known that various inflammatory cytokines are induced in peripheral tissues by psychological stress[38]. To examine the types of cytokines induced by restraint stress, we measured the expression levels of various cytokine genes by qRT-PCR from whole-body total RNA prepared after 0, 3, 5, and 10 h of restraint stress exposure (Supplementary Fig. 7a). The level of upd3 mRNA, a *Drosophila* homolog of interleukin-6 (IL-6), was most

elevated (~5.6-fold) following restraint stress treatment for 10 h (Supplementary Fig. 7a). Upd3 binds and activates the JAK-STAT receptor[39]. It was also reported that restraint stress increases the level of IL-6 mRNA in rat hypothalamus and midbrain[40]. The transcription of *spaetzle 3* (*spz3*), a gene encoding a protein that binds and activates the Toll receptor[41], was also increased by 10 h restraint stress treatment, but the degree of induction was lower (2.2-fold) than that of *upd3*. To confirm the precise expression level of upd3 and spz3 mRNA during restraint stress, we checked the expression level of these mRNAs by qRT-PCR using three biological replication samples (Fig. 6a). The expression of *upd3* and *spz3* peaked in the early phase of restraint stress (within 1.5 h), and then only upd3 increased gradually during restraint stress. Since upd3 mRNA was slightly induced under control conditions (cultivating without medium), the net degree of induction of upd3 by 10 h restraint stress treatment was ~3.4-fold, similar to spz3 mRNA (Fig. 6a). The gradual increase of *upd3* expression during restraint stress is very similar to the pattern observed for p38 phosphorylation in testes (Fig. 5e and Supplementary Fig. 5c), while spz3 mRNA was transiently induced immediately after restraint stress exposure. Thus, Upd3 appears to be a good candidate for transmitting of stress information from somatic tissue to germline tissue.

There was no difference in the upd3 mRNA level between whole body samples of wild-type and *dATF-2* homozygous mutant (Supplementary Fig. 7b), suggesting that *upd3* is not a direct target gene of dATF-2. To examine the possibility that the p38-dATF-2 pathway is involved downstream of upd3, we analyzed *Turandot A* (*TotA*), which encodes secreted peptides and is induced in response to several stresses[42]. A previous report indicated that the *TotA* gene is induced by septic injury in an

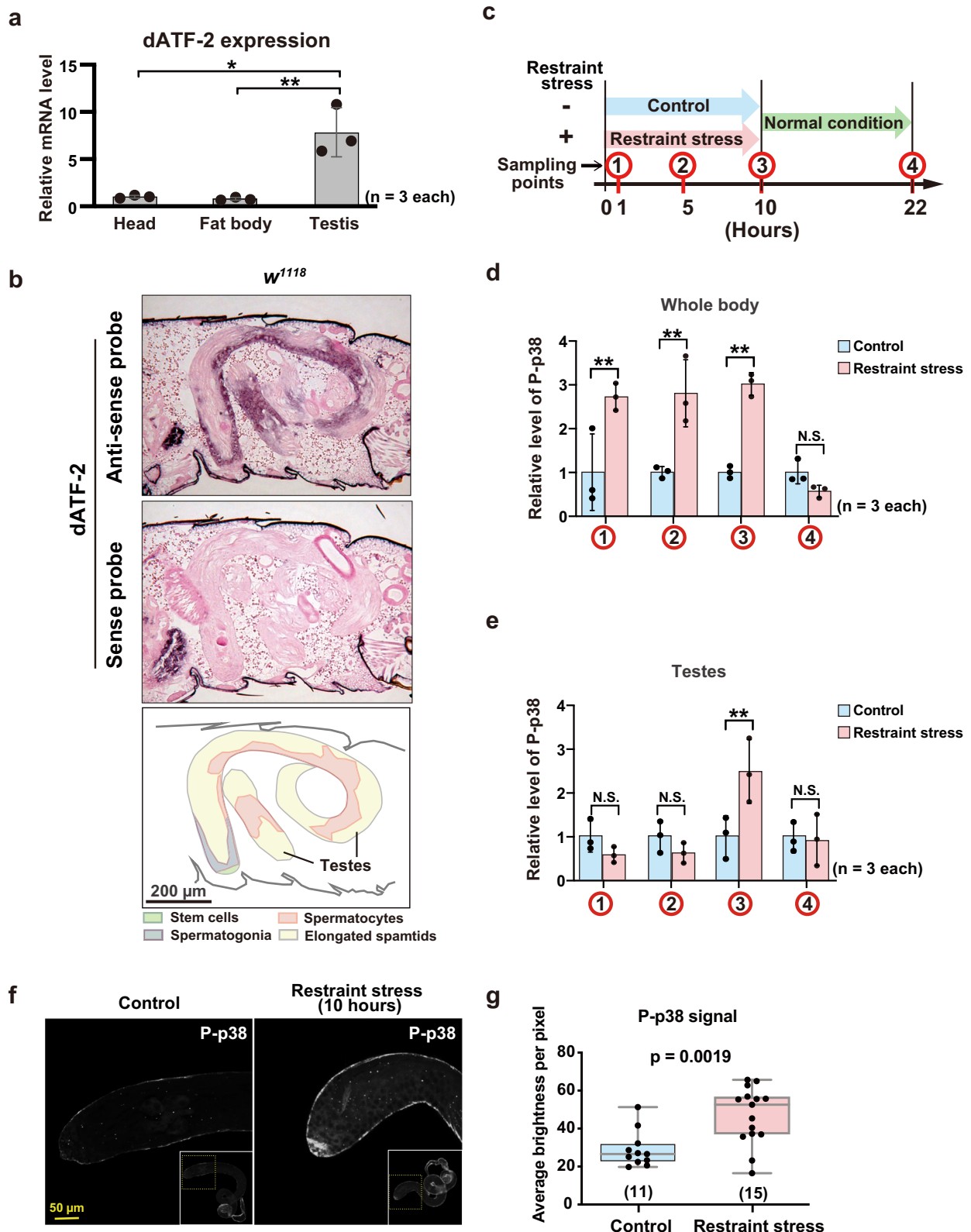

Upd3-dependent manner[43] and also regulated by MEKK1[44]. We first analyzed *TotA* expression using total RNA prepared from whole-body fly samples and observed that *TotA* expression was also induced by restraint stress (Supplementary Fig. 7c). Interestingly, the induction of TotA was observed at 12 h after 10 h restraint stress exposure, although upd3 induction was not observed at the same timepoint (Supplementary Fig. 7d). The

maintenance of TotA induction may therefore be a result of epigenetic regulation caused by restraint stress. We next tested the role of the MEKK1-dATF-2 signaling pathway in TotA induction by restraint stress. The *dATF-2^{KO}* mutant displayed a tendency of higher *TotA* expression than wild-type flies, although a clear significance was not observed (Supplementary Fig. 7e), suggesting that *TotA* is a target of dATF-2 and is suppressed by

**Fig. 5 Restraint stress induces phosphorylation of p38 in testicular germ cells. a** Quantitative real-time PCR analysis of gene expression in different tissues. Averages with s.d. are shown ($n = 3$ each). **b** RNA in situ hybridization using dATF-2 mRNA antisense (upper image) and sense (center image) probes. The schematic diagram shows the developmental stages of spermatogenesis (lower diagram). **c–e** Western blot analysis was performed using samples from $w^{1118}$ adult whole-body and testes with anti-P-p38 antibody. **c** The diagram represents the timing of sampling for the western blots. Samples were prepared at each timepoint (restraint stress for 1, 5, and 10 h and at 12 h after the 10 h restraint stress treatment). Quantitation of the P-p38 from whole-body (**d**) and testes (**e**). α-tubulin was used as an internal control. The data represent the mean ± s.d. ($n = 3$ each) with $P$ values from Student's unpaired $t$ test: **$p < 0.01$; N.S., no significant difference. The blotting data were shown in Supplementary Fig. 5b, c. **f** Immunohistochemical (IHC) staining of P-p38 in testes with/without restraint stress for 10 h. **g** Representation of average brightness per pixel from the P-p38 signal in testes samples with/without restraint stress. See also Supplementary Figs. 5 and 6.

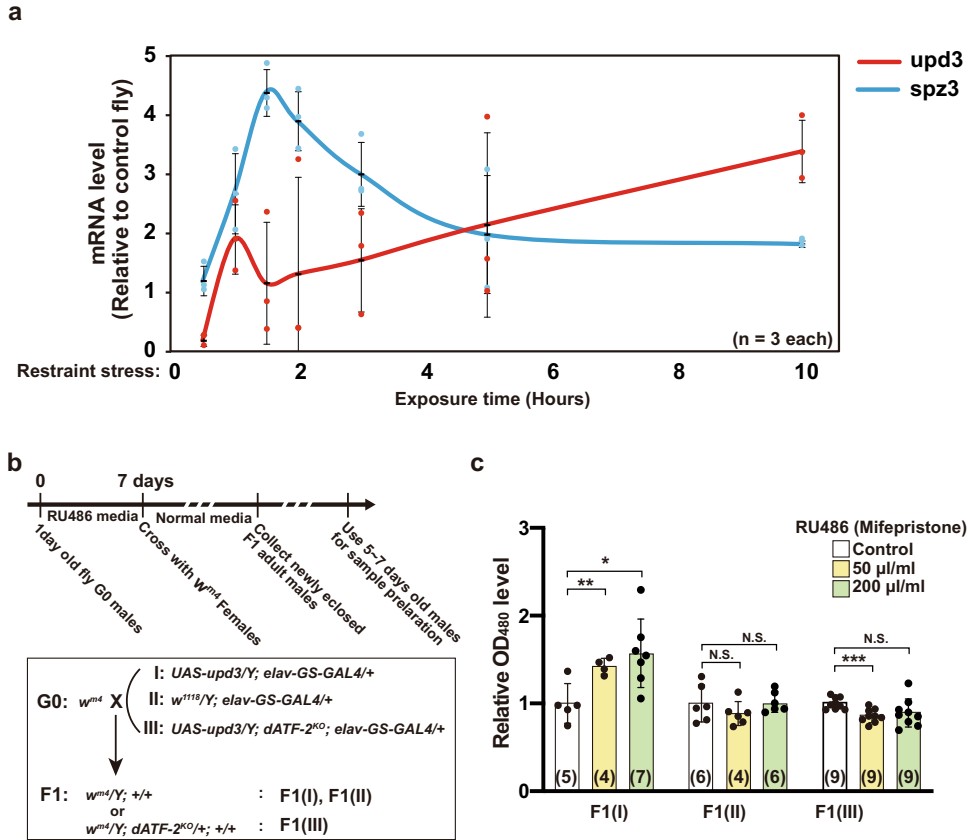

**Fig. 6 Induction of upd3 by restraint stress, and heterochromatin disruption by paternal overexpression of upd3 in neuronal cells. a** Time course of *upd3* and *spz3* gene expression during restraint stress exposure for 10 h. The value of $y$-axis represents the ratio of mRNA levels of restraint stress-treated flies to that with control flies. Averages with s.d. are shown ($n = 3$ each). Note that the data are slightly different from that in Supplementary Fig. 7a, because the results were obtained by three biological replicative samples. **b** Schematic diagram of the experimental design. Newly eclosed virgin males of each genotype (I, *upd3*-inducible; II, *upd3*-non-inducible control; III, *upd3*-inducible *dATF-2* mutant) were reared in RU486-containing medium and crossed with $w^{m4}$ females. **c** Red-eye pigment levels in F1 males (I, II, and III) were then measured (right panel). Only F1(I) exhibited an RU486-dependent increase in red-eye pigment. Averages with s.d. are shown (***$p < 0.001$; **$p < 0.01$; *$p < 0.05$; N.S., no significant difference, Student's unpaired $t$ test). Number of samples analyzed indicated in parentheses on the graph. See also Supplementary Fig. 7.

dATF-2. Furthermore, restraint stress-dependent TotA induction was severely abrogated in the *MEKK1*-deficient mutant. These results indicate that Upd3-dependent TotA induction by restraint stress requires the MEKK1-dATF-2 signaling pathway, and they suggest that the *upd3* expression pattern tightly reflects the activation timing pattern of p38 during restraint stress exposure (Supplementary Fig. 7f).

To test whether the induction of upd3 in male flies also disrupts heterochromatin and enhances *w* gene expression in offspring in PEV assays like restraint stress, upd3 was artificially overexpressed in neuronal cells using the drug-inducible pan-neuronal *elav-GS-Gal4* driver[45]. Treatment of F0 male flies with increasing amounts of the RU486 drug enhanced *w* gene

expression in offspring (Fig. 6b, c). Moreover, drug-induced *w* gene expression in F1 was not observed when F0 *dATF-2^{KO}* male flies were used (Fig. 6b, c).

Altogether, these findings suggest that the upd3 induction has the role of a transmitter of the restraint stress information which causes the dATF-2 pathway-dependent epigenetic change.

## Discussion

In the present study, we indicate that paternal restraint stress induces epigenetic status of offsprings via a p38-MEKK1-dATF-2 pathway and restraint stress-induced Upd3 might have a role in transmission of the stress information from somatic to germline

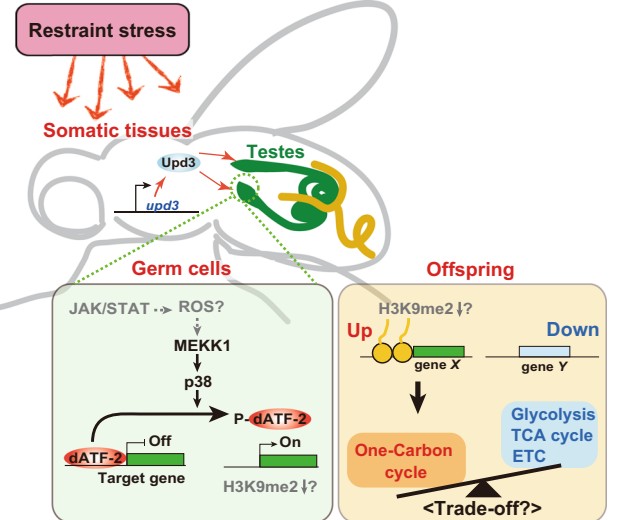

**Fig. 7 Schematic representation of the molecular mechanism of restraint stress-induced epigenetic inheritance.** Restraint stress information, received by the central nervous system (CNS) and/or other sensing tissues, promotes expression of *upd3* in peripheral somatic tissues. In germ cells, accumulated humoral Upd3 activates the JAK/STAT pathway, which subsequently activates the p38-dATF-2 pathway. Phosphorylated dATF-2 may be released from the promotor regions of its target genes, resulting in decreased H3K9me2 levels. Epigenetic marks may be retained in mature sperm and inherited by offspring. After fertilization, histone marks may act as inheritable epigenetic memory and regulate gene expression. We observed that genes involved in the one-carbon metabolic pathway were upregulated in offspring from the paternal restraint stress-exposed fathers, while genes involved in the respiratory metabolic pathway were downregulated. We assume that paternal restraint stress-induced- and dATF-2-mediated upregulation of the one-carbon cycle induces downregulation of respiratory metabolism due to a trade-off relationship between these two metabolic processes.

cells. Notably, we observed that paternal restraint stress reduces energy metabolism activity in offspring and sensitizes to rotenone toxicity. A schematic model based on our findings is shown in Fig. 7.

The stress response, orchestrated by the hypothalamic–pituitary–adrenal (HPA) axis and sympathetic nervous system via stress hormones (adrenaline, noradrenaline, and cortisol), plays a key role in physiology and in several diseases including depression[46]. Accumulated evidence has also implicated inflammatory cytokines in depression[47]. A peripheral infection can induce sickness behaviors, including depression-like behavior, which is caused by the immune-to-brain communication via proinflammatory cytokines produced from innate immune cells[48]. Previous meta-analytic reports indicate that depressed patients possess significantly higher concentrations of IL-6[49,50]. It was also reported that restraint stress activates the HPA axis and induces higher levels of IL-6 in a rodent model[51,52]. Furthermore, IL-6 knockout mice exhibit resistance to stress-induced depression-like behaviors, suggesting that IL-6 plays a key role in eliciting depression[53]. Our present results suggest that *upd3*, the *Drosophila* homolog of *IL-6*, is also induced by restraint stress. Studies on the inflammatory status of the human HMC-1 cell line showed that adrenaline enhances *IL-6*, *IL-8*, and *IL-13* production, mediated in part by the p38 signaling pathway[54]. The *Drosophila* model of myeloproliferative neoplasm indicates that p38 pathway contributes to feed-forward loop in JAK pathway by regulating *upd3* gene expression[55]. In *Drosophila*, restraint stress-dependent

*upd-3* induction in peripheral tissue may also be mediated by the p38 pathway in response to stress hormones such as adrenaline, as occurs in mammalian systems.

In the present study, we showed that Upd3 has a role in transmitting restraint stress information to germ cells. We observed that overexpression of *upd3* in neuronal somatic tissues affected the heterochromatic status of offspring in a dATF-2 dependent manner. These results indicate that humoral Upd3, secreted from somatic cells, may be able to affect the epigenetic status of germ cells via the dATF-2 pathway. Moreover, p38 can be activated in response to several extracellular factors, such as TNF, IL-1, growth factors, insulin, salt concentration, and reactive oxygen species (ROS), either directly or indirectly[56]. Thus, humoral components can act as key players to regulate the epigenetic status of germline cells. Upd3, a JAK ligand, is expressed in somatic gonads and activates the JAK/STAT pathway in male germ cells[57] suggesting that testicular germ cells are responsive to the restraint stress-induced increase in Upd3. The activation of p38 by JAK signaling has been demonstrated in the *Drosophila* immune system[58], as well as in various vertebrate systems[59]. Somatic tissue-specific overexpression of *upd3* clearly alters the heterochromatin status in the next generation. *dATF-2* is highly expressed in germ cells in testes, and expression of *TotA*, a known Upd3 signal target gene, is regulated by the MEKK1-dATF-2 pathway. Therefore, restraint stress-induced humoral Upd3 in somatic tissues may control the epigenetic status of germ cells via the p38-dATF-2 signaling pathway in *Drosophila*. Interestingly, restraint stress-dependent activation of p38 in testes was mainly induced during the latter stages of restraint stress treatment for 10 h. A relatively long time may therefore be necessary to accumulate sufficient humoral Upd3 for activation of p38 in germ cells.

In restraint stress experiments, animals may also receive mechanical stress, and it may be difficult to completely differentiate this from restraint stress. The relationship between restraint stress and mechanical stress is poorly understood. Mechanical stress induces different sets of cytokines in different types of tissues and IL-6 is one of the major components induced by mechanical stress in mammals[60]. In *Drosophila*, a recent report showed that forceps squeezing of larvae induces *upd3* gene expression[61]. IL-6 induction by either restraint stress or mechanical stress may therefore reflect a common mechanism for sensing and communicating stress information between restraint stress in the nervous system and mechanical stress in the peripheral tissues.

In *Drosophila*, the transit time from spermatagonial stem cells to mature sperm available for insemination at 25 °C is ~260 h[62]. During spermatogenesis, transcription mainly occurs in spermatagonial stem cells and spermatocytes. Herein, when fathers were exposed to restraint stress only once, only the first brood, but not successive broods, exhibited paternal restraint stress-induced heterochromatin disruption. However, exposure to restraint stress three times induced heterochromatin disruption not only in the first brood but also in successive broods. These results suggest that single restraint stress exposure induces heterochromatin disruption only in spermatocytes, whereas repeated restraint stress exposure also causes heterochromatin disruption in spermatagonial stem cells. Interestingly, Bozler et al. indicated that long-term memory is required for the maintained effect of the maternal transmission of ethanol preference after 24 h of wasp exposure[19,63]. There is a possibility that the establishment of neuronal memory by restraint stress may be important for the stable transmission of the epigenetic status in germ cells. We need further analysis to elucidate the mechanistic aspect of maintenance of epigenetic change induced by restraint stress in germ cells.

Increasing evidence indicates that the paternal environment affects the transcriptome and epigenetic status of their offsprings in several animal models[6,8,22]. Although the mechanism of paternal inheritance of epigenetic status remains unclear, some evidence suggests that histone modification is an important player[10]. Our recent study indicated that histone replacement-completed sperm retain histone H3 mainly on specific promoter regions in mice[17]. In *Drosophila*, epiallelic status of H3K27me3 can be dominantly transmitted to the progeny[24]. *Drosophila* paternal Cid/Cenp-A, a centromere histone protein, can be transmitted to progeny embryos[64]. Similarly, human spermatozoa retain and transmit nucleosomes with constitutive heterochromatin containing H3K9me3 to embryos[65]. Our previous report showed that dATF-2 is required for the establishment and maintenance of heterochromatin by regulating the H3K9me2 level, and heat shock stress induces dATF-2 phosphorylation by p38 and its release from heterochromatin, and disrupts the heterochromatin status in germ cells[8]. This causes heterochromatin disruption, which is inherited by the next generation. Based on these observations, it is likely that restraint stress disrupts heterochromatin by decreasing H3K9me2 levels in testicular germ cells, and this is transmitted to offspring. Via a similar mechanism, restraint stress may reduce H3K9me2 on dATF-2 target genes, and this may be transmitted to offspring to modulate metabolism. However, we could not analyze dATF-2 target genes in testicular germ cells by ChIP because our anti-dATF-2 antibodies worked only on embryo chromatin but not testis chromatin. This could be due to the presence of non-specific proteins that bind to anti-dATF-2 antibodies in testes. To detect epigenetic effect of paternal restraint stress, we crossed restraint stress-treated $w^{1118}$ males with $w^{m4}$ females. We observed the level of histone H3K9me2 was slightly but significantly decreased in the *w* gene locus of $w^{m4}$ offsprings from restraint stress-treated fathers. Although the X chromosome harboring the $w^{m4}$ gene in the male offspring was derived from the unstressed female, paternal restraint stress increased *w* expression in male offspring, which is reminiscent of paramutation. We and another group have reported that paternal stress can influence epigenetic status on maternally supplied X chromosome using the $w^{m4}$ strain[6,8]. A previous report also indicated that quantitative content of heterochromatin on the Y chromosome affects PEV phenotypes of $w^{m4}$ flies[66]. Physical pairing between the heterochromatic regions of X chromosomes and other chromosomes, or the *trans*-action of some molecules, such as noncoding RNA, may induce partial disruption of the heterochromatin on the X chromosome. Small noncoding RNAs have been also reported as a factor of trans-generational transmission of paternal phenotypes in several model animals[67]. Noncoding RNAs in sperm may also influence the transgenerational effect by paternal restraint stress. Further analysis is required to elucidate the detailed mechanism by which restraint stress affects metabolism in offspring via dATF-2 in testicular germ cells.

Consistent with a previous report on mice[29], the present study indicates that paternal restraint stress affects metabolism in offspring, albeit differently, possibly due to differences among species and experimental conditions. In the present study, paternal restraint stress enhanced the expression of genes involved in one-carbon metabolism, while paternal restraint stress decreased the expression of genes involved in glycolysis and the TCA cycle in the previous work on mice. Restraint stress disrupts heterochromatin possibly by decreasing the H3K9me2 level, which is thought to directly enhance the transcription of dATF-2 target genes. Therefore, downregulation of genes by paternal restraint stress, such as glycolysis-related genes, may occur indirectly via upregulation of one-carbon metabolism-associated genes. The *phosphoglycerate dehydrogenase* (*PHGDH*) gene is important in balancing glycolysis and one-carbon metabolism, and its amplification in some cancer cells causes the diversion of a relatively large amount of glycolytic carbon into serine and glycine metabolism[68]. The upregulation of one-carbon metabolism genes, including *PHGDH*, enhances the levels of glutathione and taurine, which are major factors for reducing ROS during detoxification[69,70]. ROS stabilizes the transcription factor hypoxia-inducible factor-1α (HIF-1α)[71], which activates genes related to glycolysis[72]. Therefore, one possibility is that indirect downregulation of glycolysis genes may be due to decreased activity of HIF-1 following the lowering of ROS levels by paternal restraint stress.

Multiple studies have shown that psychological stress affects metabolism[73], and metabolism conversely modulates the response to psychological stress[74]. The incidence of both depression and diabetes increases with age, and inflammatory cytokines play a key role in eliciting both diseases. Growing evidence indicates that depression and type 2 diabetes share biological origins, particularly overactivation of a cytokine-mediated inflammatory response, potentially through dysregulation of the HPA axis[75]. Thus, psychological stress and metabolism are tightly connected, and our study indicates that psychological stress modulates metabolism in offspring through intergenerational inheritance via sperm. Some previous evidence showed that diet composition may attenuate stress-induced symptoms[76], and the present results, including precise gene expression changes induced by paternal restraint stress, may contribute to the development of useful foods or supplements with therapeutic benefits.

## Methods

**Drosophila stocks**. All flies were maintained at 25 °C on standard medium. The strains used in this study were $w^{1118}$ (wild-type; BDSC 5905), In(1) $w^{m4}$ ($w^{m4}$ for short), $Mekk1^{Ur36}$[77], elav-GS-GAL4 (BDSC 43642), and UAS-upd3 (Harvard EXELIXIS stock collection P[XP]d04951)[78]. To generate the $dATF-2^{KO}$ null mutant, the ends-out targeting approach was used[34]. The $dATF-2^{KO}$ mutant lacks the entire dATF-2 gene region and does not express dATF-2 (Supplementary Fig. 2). To normalize the genetic background, $w^{m4}$ and $dATF-2^{KO}$ were back-crossed six times with $w^{1118}$.

**Standard fly medium**. We used the following reagents for standard fly food: agar powder: WAKO 010-15815 (500 g), D(+), Glucose: WAKO 045-31167 (10 kg), Dry yeast: Beer Yeast Korea Inc. Dry Yeast G2 (20 kg), and Cornmeal: SUNNY MAIZE CO., LTD. Corn grits No. 4 M (25 kg). After entirely dissolving all reagents (21 g of agar powder, 300 g D(+)-Glucose, 150 g of dry yeast, 210 g of cornmeal) in 3.5 l Mili-Q water with heating, the mixture is autoclaved and cool down up to 80 °C. Then, 10 ml of propionic acid is added and dispense the medium into each vial (5 ml/vial) and plugging.

**Restraint stress**. To induce restraint stress, 1-day-old adult males (separated from females on day 0, 20 flies/vial) were anesthetized and gently placed between two soft sponge plugs (COW 30t × 27ø; HIGH TECH LLC, Chiba, Japan) in a plastic vial (MKC-30 [small]; HIGH TECH), as shown in Supplementary Fig. 1a, b. Flies were exposed to restraint stress for 10 h, and then transferred into a new vial containing normal medium and rested for 14 h. We repeated the cycle of restraint stress up to three times and then crossed with appropriate females. To eliminate the effects of differences in genetic and rearing conditions in parents, we always compared the F1 progenies from sibling parents in each experiment. All steps were conducted under constant temperature (25 °C) and humidity (60%) with 12 h light:12 h dark cycle.

**PEV assay and pigment quantification**. Eclosed flies were separated by sex and reared in new vials (30 flies/vial) at 25 °C, and flies at 5−6 days old were transferred into 1.5 ml tubes, frozen in liquid nitrogen, and stored at −30 °C until use. Frozen flies were decapitated by vigorous shaking and vortexing, and ten heads were homogenized with a pellet mixer in 250 µl methanol/0.1% HCl, followed by overnight extraction at 4 °C. After centrifugation at 15,000 × *g* for 5 min, the absorbance of each sample was measured at 480 nm using a DU 730 spectrophotometer (Beckman Coulter, CA, USA).

**Paternal overexpression of upd3 in neuronal cells**. To overexpress upd3 specifically in adult neuronal cells, we used RU486-inducible elav-gene-switch-GAL4

system[50]. We prepared RU486 medium mixed with RU486 from a 25 mg/ml stock solution in 100% ethanol to a final concentration of 50 and 200 μg/ml. As a control medium, we used standard medium with 100% ethanol only. Schematic diagram of the experimental procedure are described in Fig. 6b. One-day-old F0 males are reared in the RU486-containing medium for a week, and then cross with wm4 females in a vial with normal medium. After eclosion of F1 males, we measured the red-eye pigment level of 5–7 days old F1 males.

**Chromatin immunoprecipitation (ChIP)**. Twenty adult males were collected and homogenized with ~15 strokes of a loose Dounce pestle in 5 ml of cold M3 *Drosophila* medium. After centrifugation (300 × *g*, 1 min), The supernatant was transferred into a new tube and washed with M3 *Drosophila* medium and then phosphate-buffered saline (PBS) once. The cells were pelleted by centrifugation (1000 × *g*, 5 min), and then resuspended by pipetting 1 ml of the cell-burst buffer (25 mM HEPES-KOH [pH 7.3], 1.5 mM MgCl₂, 10 mM KCl, 0.2% NP-40 and proteinase inhibitor cocktail). Then, cells were washed with 1 ml of fixing buffer (25 mM HEPES-KOH [pH 7.3], 1.5 mM MgCl2, 10 mM KCl, 0.1% NP-40 and proteinase inhibitor cocktail) and resuspend in 1 ml of fixing buffer. Then, 62.5 μl of 16% paraformaldehyde solution was added and incubate at RT for 5 min. The cross-linking reaction was stopped by adding 62.5 μl of 2 M glycine. After washing with 1 ml of fixing buffer twice, cells were collected by centrifugation (2000 × *g*, 2 min) and frozen with nitrogen liquid. Next, the pellet was resuspended in 300 μl of nuclear lysis buffer (50 mM Tris-HCl [pH 8.0], 10 mM EDTA, 1% sodium dodecyl sulfate (SDS) and proteinase inhibitor cocktail) and incubated on ice for 20 min. The samples were sheared by sonication with Q800R sonicator (Qsonica, CT), and the sonication conditions were 15 s pulse on, 45 s pulse off, and 30 W of power for 2 min. After centrifugation (15,000 × *g*, 15 min), the supernatant was diluted with 2.9 ml of IP dilution buffer (20 mM Tris-HCl [pH 8.0], 150 mM NaCl, 2 mM EDTA, 0.5% Triton X-100 and proteinase inhibitor cocktail). For immunoprecipitation, samples (1 ml each) were incubated with antibodies prebound to protein A or G Dynabeads (Invitrogen, Carlsbad, CA). After overnight incubation at 4 °C, the beads were washed three times with wash buffer I (0.5% Triton X-100, 0.1% SDS, 2 mM EDTA, 20 mM Tris-HCl [pH 8.0], 150 mM NaCl and proteinase inhibitor cocktail) and once with wash buffer II (0.5% Triton X-100, 0.1% SDS, 2 mM EDTA, 20 mM Tris-HCl [pH 8.0], 500 mM NaCl and proteinase inhibitor cocktail). The beads were washed with wash buffer III (0.5% NP-40, 1 mM EDTA, 20 mM Tris-HCl [pH 8.0], 0.25 M LiCl and proteinase inhibitor cocktail) and twice with TE. The DNA samples were eluted with 100 μl elution buffer (TE buffer containing 1% SDS) and voltexed for 10 min. Then the samples were transferred to new tubes, mixed with 100 μl TE containing RNase A, and incubated at 65 °C for overnight for reverse cross-linking. DNA samples were purified and resuspended in 100 μl of TE using the QIAquick PCR purification kit (QIAGEN, Hilden, Germany). DNA levels were determined by qPCR on a QantStudio 3 Real-Time PCR system (Applied Biosystems). Antihistone H3 (ab1791; Abcam, Cambridge, UK), and antihistone H3K9me2 (ab1220; Abcam, Cambridge, UK) were used for ChIP assay. The primer pairs were used as described in ref. [33].

**Quantitative RT-PCR**. To obtain total RNA from various body parts (head, fat body, and testes), five flies were dissected in RNAlater (Qiagen, Hilden, Germany) on ice, and total RNA was extracted using Isogen II (Nippon Gene, Tokyo, Japan). To evaluate cytokines and TotA induction by restraint stress, whole-body samples from five males were used for total RNA extraction, and qRT-PCR was performed using a One Step SYBR PrimeScript PLUS RT-PCR kit (Takara Bio, Shiga, Japan) and an Applied Biosystems ABI Prism 7000 Sequence Detection System. Sequences of PCR primers are listed in Supplementary Table 2.

**RNA in situ hybridization (ISH)**. A 618 bp DNA fragment corresponding to the nucleotide positions 646–1263 of *dATF-2* was subcloned into the pGEMT-Easy vector (Promega, Madison, WI, USA) and was used for generation of sense or antisense RNA probes.

Paraffin-embedded fly sections (6 μm) were obtained from Genostaff Co., Ltd. (Tokyo, Japan) Adult males were fixed with Tissue Fixative (Genostaff Co., Ltd.), embedded in paraffin using standard procedures, and cut into 6-μm sections. For ISH, fly sections were deparaffinized with xylene and rehydrated through a graded ethanol series in PBS. Sections were fixed in 4% paraformaldehyde in PBS for 15 min, washed with PBS, treated with 6 μg/ml proteinase K in PBS for 30 min at 37 °C, washed with PBS, refixed with 4% paraformaldehyde in PBS, washed again with PBS, and then placed in 0.1 M triethanolamine HCl (pH 8.0) containing 0.25% acetic anhydride for 10 min. After washing with PBS, sections were dehydrated through a graded ethanol series. Hybridization was performed with probes at concentrations of 3000 ng/ml in Probe Diluent-1 (RPD-01; Genostaff Co., Ltd.) at 60 °C for 16 h. After hybridization, sections were washed in 5× HybriWash (SHW-01; Genostaff Co., Ltd.) and an equal volume of 5× SSC at 60 °C for 20 min, and then in 50% formamide in 2× HybriWash at 60 °C for 20 min, followed by RNase treatment in 50 μg/ml RNase A in 10 mM Tris-HCl (pH 8.0) containing 1 M NaCl and 1 mM EDTA for 30 min at 37 °C. Sections were washed twice with 2× HybriWash at 60 °C for 20 min, twice with 0.2× HybriWash at 60 °C for 20 min, and once with PBST (0.1% Tween-20 in PBS). After treatment with G-block (GB-01; Genostaff Co., Ltd.) for 30 min, sections were incubated with anti-DIG AP

conjugate (Roche) diluted 1:1000 with TBST for 2 h at room temperature (RT). Sections were washed twice with TBST and then incubated in 100 mM NaCl, 50 mM MgCl₂, 0.1% Tween-20, and 100 mM Tris-HCl (pH 9.5). Coloring reaction was performed with NBT/BCIP solution (Sigma) overnight, followed by washing with PBS. Sections were counterstained with Kernechtrot stain solution (Muto pure chemicals, Tokyo, Japan), dehydrated, and then mounted with Malinol (Muto pure chemicals).

**Western blotting**. To examine dATF-2 levels in testes of *w1118* and *dATF-2KO* males, 50 pairs of testes were dissected and homogenized with 100 μl RIPA buffer (20 mM Tris-HCl [pH 8.0], 300 mM NaCl, 1% NP-40, 0.1% sodium deoxycholate, 0.1% SDS, 1 mM EDTA, and proteinase inhibitors). Samples were incubated on ice for 20 min and centrifuged at 15,000 × *g* for 10 min. Supernatants were mixed with the same volume of 2× SDS sample buffer, boiled for 10 min, and subjected to western blotting with anti-dATF-2 antibody (1:1000), which was raised against the pseudomonas exotoxin full length dATF-2 fusion protein[79].

To detect P-p38 in testes, ten pairs of testes were dissected and homogenized with 50 μl 2× SDS sample buffer. For detecting Phospho-p38 from whole body, Five fly males were directly homogenized with 50 μl 2× SDS sample buffer. After boiling and centrifugation, supernatants were used for western blotting with rabbit phospho-p38 MAPK monoclonal antibody (1:1000; Thr180/Tyr182; catalog number 9215; Cell Signaling Technology, Danvers, MA, USA) and mouse anti-alpha-tubulin antibody (1:2000) as an internal control.

Membranes were probed with Alexa Fluor Plus 680/800 goat anti-rabbit IgG or anti-mouse IgG secondary antibodies (1:20000; Invitrogen, Carlsbad, CA). Signal bands were detected using an Odyssey Fc Imager (LI-COR, Lincoln, NE, USA).

**Immunohistochemistry of testes**. Testes from 1-day-old males with/without restraint stress were dissected with 1× testes buffer (183 mM KCl, 47 mM NaCl, 10 mM Tris-HCl [pH 6.8], protease inhibitor cocktail, and phosphatase inhibitor cocktail) and fixed with fixation buffer (4% paraformaldehyde in testes buffer) for 40 min. Samples were then washed with PBTx (PBS containing 0.5% Triton-X-100, protease inhibitor cocktail, and phosphatase inhibitor cocktail) twice, and then treated with Image-iT FX signal enhancer (Invitrogen) for 30 min at RT. After washing with PBTx, samples were blocked with 10% goat serum in PBTx for 1 h and then incubated with anti-P-p38 (1:100; catalog number 9215; Cell Signaling) overnight at 4 °C, rinsed with PBTx at RT for 10 min (three times), and then labeled with Alexa Fluor® 488 Goat Anti-Rabbit IgG (#R37116; Thermo Fisher Scientific, Waltham, MA). All images were obtained using a confocal laser scanning microscope (LSM780; Carl Zeiss, Germany) with the same parameter settings. To compare the intensity of the P-p38 signal in samples with/without restraint stress, the average brightness per pixel of testes regions was measured using Adobe Photoshop CS6 (Adobe Systems, CA, USA) and normalized against the average brightness of the outer region of testes in each image.

**RNA-seq analysis**. *w1118* or *dATF-2KO* males subjected to paternal restraint stress (three times) or controls were crossed with *w1118* females. Five-day-old F1 males derived from paternal restraint stress (three times) and control flies were vigorously shaken after freezing in liquid nitrogen and storing at −80 °C (50 flies/sample) to separate heads and other body parts. Total RNA from the resulting samples was isolated using Isogen II (Nippon Gene), and poly(A) + RNA was purified using an Oligotex-dT30 <Superå purification kit (Takara Bio). cDNA libraries (*n* = 5 per group) were generated using TruSeq Stranded Total RNA and a Ribo-Zero Gold LT Sample Prep Kit (Illumina, San Diego, CA, USA). We then produced a set of ~8.2 × 10⁷ paired-end reads (100 bp × 2, insert size 160 bp) from each cDNA library using a Hiseq 2500 instrument (Illumina). Sequenced reads were mapped to the *Drosophila melanogaster* genome (UCSC, dm3) using TopHat. The expression levels were estimated and compared using Cufflinks and Cuffdiff programs.

**Metabolome analysis**. *w1118* or *dATF-2KO* males subjected to paternal restraint stress (three times) or controls were crossed with *w1118* females. Five-day-old F1 males (30 flies) derived from paternal restraint stress (three times) and control flies were stored at −80 °C in 500 μl ethanol containing internal standards (20 μmol/l each of methionine sulfone and D-camphor-10-sulfonic acid) for metabolome analysis. After adding zirconium beads (5 mm × 2, 3 mm × 4), samples were homogenized using a Shake Master NEO (Bio Medical Science, Tokyo, Japan) for 1 min at 1500 rpm. Homogenates were mixed with 500 μl CHCl₃ and 200 μl Milli-Q water and centrifuged at 4600 × *g* for 15 min at 4 °C. A 300 μl sample of the aqueous layer was then subjected to ultrafiltration (5 kDa cut-off) by centrifugation at 9100 × *g* for 160 min at 20 °C. Samples were dissolved in 50 μl of Milli-Q water containing 200 μmol/l each of 3-aminopyrrolidine and trimesic acid before CE-TOFMS analysis, which was performed as described previously[80]. Raw data from CE-TOFMS were processed using our proprietary automatic integration software (MasterHands)[81]. CE-TOFMS analysis was performed on sic samples per group and subjected to further analysis. Statistically significant differential metabolites (*p* < 0.05, *t*-test) among groups, and their relative amounts, were portrayed using a heatmap, and PCA and metabolome Enrichment Set Analysis were conducted using a web-based application[82].

**Rotenone stress test**. To examine the effects of rotenone on the survival of individual flies, a single fly was placed in each well of a 384-well microplate containing rotenone medium (5% sucrose, 0.5% dry yeast, 0.3% propionic acid, 1.5% agar, and 5 mM rotenone). The plate was sealed with a lid (acrylic plate with holes, 0.5 mm diameter, centered above each well). The plate as scanned on a charge-coupled device flatbed scanner at 5-min intervals. Individual death times were determined at high temporal resolution (5 min) using a series of time-lapse images. We determined the death timing of flies using an in-house developed software.

**Statistics and reproducibility**. All data were analyzed by GraphPad Prism version 8. Error bars represent means ± s.d. of more than three biologically independent samples and each experiment was independently repeated more than two times with similar results unless otherwise mentioned. A student's unpaired, two-tailed $t$ test was performed to compare differences between groups in each experiment (***$p < 0.001$; **$p < 0.01$; *$p < 0.05$; n.s., no significant). A statistical analysis of the survival curves was conducted by log-rank (Mantel–Cox).

**Reporting summary**. Further information on research design is available in the Nature Research Reporting Summary linked to this article.

## Data availability

RNA-seq data are accessible in the NCBI Gene Expression Omnibus (GEO; https://www.ncbi.nlm.nih.gov/geo/) under the accession number GSE140950. Source data for the main and supplementary figures are included in Supplementary Data 1. Any other data not included in the paper or supplementary materials is available from the authors upon reasonable request.

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

## Acknowledgements

The authors would like to thank Dr Hong Yang and Bloomington stock center for sending stocks and vectors. We thank Yuko Iijima for her technical assistance. We are very grateful to Dr Yoichi Shinkai for helpful support. This work was supported by the AMED (Japan Agency for Medical Research and Development) [18gm0510015h0006 to S.I., JP18gm1110001 to K.-H.S., JP19gm1010009 to S.F.]; MEXT (Ministry of Education, Culture, Sports, Science and Technology) [16H01413 to S.I.]; JST PRESTO (Japan Science and Technology agency, Precursory Research for Embryonic Science and Technology) [JPMJPR12M5 to K.-H.S., JPMJPR1537 to S.F.]; JSPS KAKENHI (Japan Society for the Promotion of Science KAKENHI) [26440192, 19K06497 to K.-H.S., 18H04805 to S.F.]; The Takeda Science Foundation (to S.F.); and The Food Science Institute Foundation (to S.F.).

## Author contributions

K.-H.S. and S.I. conceived and designed experiments. K.-H.S., N.H.L., Y.K., N.Y., R.N., S.M., A.H., S.F., S.K., T.S., and K.S. performed experiments and analyzed data. K.-H.S. and S.I. wrote the manuscript with contribution from all authors. K.-H.S. and S.I. are co-supervising authors.

## Competing interests

The authors declare no competing interests.
