## [Peer Review File · Communications Biology]

Reviewers' comments:

Reviewer #1 (Remarks to the Author):

Recent evidence supporting existence of epigenetic inheritance in diverse animal species
including *Drosophila* and mammals has generated immense interest in revealing the
mechanisms underlying the phenomenon. Given this, the Seong et. al.'s manuscript providing
a mechanistic analysis of epigenetic inheritance in *Drosophila* is of important consideration. The
authors present evidence implicating p38-dATF-2 signaling dependent histone changes in a fly
model of restraint stress induced germline mediated paternal inheritance. The manuscript thus
relates to the two most fundamental questions concerning epigenetic inheritance - how the
information is transferred from soma to the germline, and what germline epigenetic factor is
inherited to the next generation. The work involves a rodent model inspired new fly model, and
is based on hypotheses based mostly on molecular parallels between the two species. The
results are novel, interesting, and consistent with previous findings supporting germline
mediated intergenerational effects on metabolic traits in *Drosophila*. Epigenetic inheritance is
poorly understood in mechanistic terms, with the findings reported by Seong et. al. offering
newer insights. However, the manuscript presentation requires necessary changes and editing.

Major comments:

1. Introduction is difficult to follow due, in large part, to consistent mixing of evidence from
yeast to humans, epidemiology to experimental studies, parental to paternal inheritance, and
so on. Also, it starts abruptly, without a general background of environmentally triggered
inter/transgenerational epigenetic inheritance, its potential mechanisms, epigenetic
differences between species (e.g., DNA methylation mechanisms not relevant in *Drosophila*) etc.
Given that epigenetic inheritance is a highly controversial subject, a general background is
greatly desirable. Needless to say, key references need to be cited in the context of each of the
above points.

The authors have not referred to a single *Drosophila* study on inter/transgenerational
inheritance in the Introduction, while their work is on fly! I suggest that following a general
background as above, a brief paragraph should be dedicated to environmental factor induced

epigenetic inheritance in *Drosophila*.

The last paragraph may introduce the present work, the restraint stress model, in particular.

The original rodent model and its molecular correlates should be briefly explained first, followed

by the proposed fly model and its expected molecular counterparts. The description

surrounding ATF2 is excessive and may be shortened by including only the most relevant

information for an Introduction section, with the rest accommodated in the later sections, as

appropriate. The paragraph should finally state the hypotheses that have been tested in the

study to understand molecular mechanisms.

2. Given that the authors have used a new fly model (RS model) to understand epigenetic

inheritance, it would have been more appropriate if the model was described first to plainly

demonstrate occurrence of epigenetic inheritance as a phenomenon, irrespective of detailed

hypothesis driven analysis. The last three sub-sections of results (transcriptomic, metabolomic,

and phenotype assessment) should have been therefore presented first. Rest of the

sub-sections relate to details and can follow the above three.

Minor comments:

1. Lines 1-2: "Paternal restraint stress affects metabolism in the offspring via *Drosophila* ATF-2

in the germ cells" – may be changed to "Paternal restraint stress affects offspring metabolism

via ATF-2 dependent mechanisms in the germ cells in *Drosophila*"

2. Line 65: space before comma!

3. Line 70: space before comma!

4. Lines 93-94: "Injection of TNF- α into F0 male mice, known to be.." should be reframed as "In

F0 male mice, injection of TNF- α , known to be.."

5. Line 95: ref 23 should be cited

6. Line 132: "resuspend 1ml", "in" is missing

7. Line 155: "and" is missing before anti-histone

8. Lines 259-261: it can simply be stated that an in-house developed software was used...

instead of referring to a specific software that is unpublished

9. Lines 265-304: Figure 1A is followed by Figure 1C in the text, with Figure B missing?

10. Lines 265-304: Text does not clearly indicate the tissue of interest (testicular germ cells).

Please mention as appropriate.

11. Lines 289-290: "1st brood but also in the 2nd and 4th broods" – as 3rd brood shows
insignificant but higher expression, it may be indicated in the text

12. Lines 295-296: "an rDNA coding region which located in a heterochromatic region" – 'is'
missing after 'which'

13. Line 301: "can induces" – can induce

14. Lines 311-315: "We observed the F1 offspring from the dATF-2 mutant exhibited high level
of w gene expression as described previously (19), and pRS did not further affect the level of w
gene expression (Figure 1G,..... Figure 1H": But the figure legend shows eye pigment levels
and not w gene expression for G and H! Also, it is unclear what "control" means in these figure
panels. Where has it been shown that dATF-2 exhibit higher level of phenotypic trait? Reference
to previous work, that too in relation to a different mutant type, further compounds the problem.
The authors need to clarify all these issues.

15. Lines 348-350: "we have checked the expression level of these mRNAs by qRT-PCR using
three biological replication samples (Figure 3B). We observed both upd3 and spz3 made peaks"
– language correction required, e.g., we checked.....the expression made peaks is odd

16. Line 384: "RS information which involved in dATF-2 pathway-dependent epigenetic change",
please correct the language

17. Line 387: "because we wanted to analyze under normal chromatin status" – please correct
the language

18. Line 431: "which is involved glycolysis" – language correction required

19. Line 510: "repeated exposure to RS three times" – repeated as well as three times!
Language correction needed

20. Line 540: whichis – space required

21. Line 585: "Supplementary Data are available at NAR online" – an example of poor editing

22. Supplementary Figure 1 legend: the last sentence is confusing. Please state clearly that the
same refers to previous work.

23. Line 632: stress instead of dtress

24. Line 549: "A sperm non-coding RNAs may also" – language correction needed

25. Lines 852-853: Figure 1 legend for panels F, G, and H are wrongly indicated.

26. Line 857: Figure 2 legend: sense and anti-sense probes are not matching with labelling
shown in the figure (lower and upper images)

27. Line 871: "No that" - Note that

- 28. Figure 2: Color scheme for spermatogonia does not seem to match
29. Figure 3: Figure 3A may be accommodate in Supplementary Figure 5 as it is a single sample
based preliminary screening.
30. Figure 4D: this panel seem unnecessary, given supplementary figure 6.
31. Figure 7: it seems unnecessary
32. The term "father flies", used throughout the manuscript, is odd and may be replaced by
fathers or male parents etc.

Reviewer #2 (Remarks to the Author):

Seong et. al present their work, entitled, "Paternal restraint stress affects metabolism in the
offspring via Drosophila ATF-2 in the germ cells" In this study, the authors investigate the role
of environmental stressors on the paternal germline and its subsequent effect on F1 Drosophila
offspring. What is very exciting is that this phenotype was not passed onto the F2, but rather,
halted in the F1 state. Interestingly, the stress causes heterochromatic changes in the paternal
genome leading to gene expression changes in the offspring. The key epigenetic factor shown
to be involved is ATF-2, as shown by mutant analysis. Collectively, this body of work identifies
paternal stress induces metabolic changes in offspring that have not experienced the stress,
demonstrating environmental stressors induce epigenetic changes yielding phenotypic
outcomes. This is a truly remarkable observation, and is important to the emerging field of
germline physiology, epigenetics, and transgenerational inheritance. I believe that this
manuscript will draw broad interest from multiple research fields, including neuro science,
epigenetics, evolution and ecology, in addition to being of interest to the public.

It is my recommendation that this important manuscript be accepted pending essential
revisions outlined below:

Major Comments

The authors provide an extremely important body of work. I also wish to commend the authors
on the logical presentation of the manuscript, as well as clear and concise figures. The models
are especially helpful! Thank you as well for Figure 7 (model).

However, I have a few concerns on the genetic basis of the paternal transgenerational effects:

• “However, the following three successive broods did not exhibit higher w gene expression.
Exposure to RS twice also yielded similar results (Figure 1D). Repetitive RS exposure three
140 times resulted in higher w gene expression not only in the 1st brood but also in the 2nd and 4th
broods (Figure 1E).”

o This is an especially amazing finding by the authors and needs to be further analyzed. One
might predict that long term memory of the stress has created the maintained inheritance of
additional brood, as shown in other *Drosophila* transgenerational studies. I believe it would be
important to test some learning and memory genes (i.e. orb2) to see how similar or different
this paradigm is. Ideally, a inducible line would be used to knockdown the gene in the father but
allow for wild-type levels in offspring. This experiment would also provide evidence to author
speculation of the nervous system communicating to the germline.

• The RU486 feeding protocol is not described in the methods (i.e. how much was fed, how
often, how was it administered, etc.). I am also unable to find the solvent only control for Figure
3C, the ideal control (rather than 0 microliter/mL).

• I am fascinated by the observation that the F1 male and female flies have inherited changes,
but the F2 do not. I would very much like to see if the effects are additive or an ‘all or nothing’
result. For example, placing the F1 males into the sandwich and then testing the F2’s. Is the
phenotype enhanced or the same? I believe that this is a critical question. Either result will place
this manuscript on a unique level as being the only manuscript that I am aware of for examining
trans and multi generational effects in *Drosophila* as a function of paternal stress. Additive
results suggest that further epigenetic changes take place (perhaps further enhancing
metabolites in F2). The same result suggests an ‘all or nothing’ response, which is a curious but
exciting evolutionary finding.

Collectively, if the above points are addressed, this paper will become the seminal work on
paternal stress affecting germline physiology and transgenerational effects in F1 findings in
*Drosophila*.

Minor Comments

• Line 102: "standard Drosophila media"—please indicate recipe/citation for this

• Methods: what temperature/humidity/light-cycle is the resistant stress performed at?

• Line 110: "virgin males"—males are ready to mate upon eclosion, so unless they were
removed upon eclosion from vials, it is very difficult to make this claim. Consider "day 0" males
as an alternative.

• PEV assay: Where male and female flies used in the "30 flies"?

• Line 127: please change "wash" to "washed"

• Line 150: "Ten" needs to be lower case to "ten"

• Line 260: "sapphire software"—if possible, please elaborate, even if just a little, on how the
software works here.

• Line 269: "RS stress"—please consider including a statement on the physiological relevance of
the fly sandwich—i.e. how/when/why might this stressor happen in nature (if at all) then how
this stress resembles other stressors.

• Figure 1 F demonstrates changes in H3K9me2. The authors find that the F2 generation is not
affected why way of variegation. Consider testing the H3K9me2 state in F2—perhaps there is
still a marginal change, but not sufficient to induce changes. If there is a change, this could
demonstrate a 'primed' state, where stressing the F2 could cause a stronger effect than the F0.

• Line 387: incomplete sentence/ subject verb disagreement

• Discussion: Consider discussing other Drosophila transgenerational studies in the discussion
and highlighting the unique finding in this manuscript.

o J Hered. 2019 May 7;110(3):300-309. doi: 10.1093/jhered/esz009.

♣ Age of Both Parents Influences Reproduction and Egg Dumping Behavior in Drosophila
melanogaster.

♣ Mossman JA1, Mabeza RMS1, Blake E1, Mehta N1, Rand DM1.

o Elife 2018

♣ Transgenerational dynamics of rDNA copy number in Drosophila male germline stem cells

♣ Kevin L Lu, Jonathan O Nelson, George J Watase, Natalie Warsinger-Pepe, Yukiko M
Yamashita University of Michigan, United States

o Bozler J (2019) Maternal Priming of Offspring Immune System in Drosophila. G3: Genes,
Genomes, Genetics. doi: doi.org/10.1534/g3.119.400852. PMID:

o Bozler J (2019) Transgenerational inheritance of ethanol preference is caused by maternal NPF
repression eLife. doi: doi.org/10.7554/eLife.45391. PMID: 31287057

Balint Z. Kacsoh | Ph.D.

Response to the reviewers' comments

We really appreciate the reviewers' comments which are helpful to improve the quality of the manuscript. Our reply to each comment is shown below.

Reviewer's text will be in blue whereas our part will be in black.

Response to comments by reviewer #1

Major Comments:

1. Introduction is difficult to follow due, in large part, to consistent mixing of evidence from yeast to humans, epidemiology to experimental studies, parental to paternal inheritance, and so on. Also, it starts abruptly, without a general background of environmentally triggered inter/transgenerational epigenetic inheritance, its potential mechanisms, epigenetic differences between species (e.g., DNA methylation mechanisms not relevant in *Drosophila*) etc. Given that epigenetic inheritance is a highly controversial subject, a general background is greatly desirable. Needless to say, key references need to be cited in the context of each of the above points.

The authors have not referred to a single *Drosophila* study on inter/transgenerational inheritance in the Introduction, while their work is on fly! I suggest that following a general background as above, a brief paragraph should be dedicated to environmental factor induced epigenetic inheritance in *Drosophila*.

The last paragraph may introduce the present work, the restraint stress model, in particular. The original rodent model and its molecular correlates should be briefly explained first, followed by the proposed fly model and its expected molecular counterparts. The description surrounding ATF2 is excessive and may be shortened by including only the most relevant information for an Introduction section, with the rest accommodated in the later sections, as appropriate. The paragraph should finally state the hypotheses that have been tested in the study to understand molecular mechanisms.

Reply:

We appreciate this comment. We have changed the Introduction as the reviewer recommended.

2. Given that the authors have used a new fly model (RS model) to understand epigenetic inheritance, it would have been more appropriate if the model was described first to plainly demonstrate occurrence of epigenetic inheritance as a phenomenon, irrespective of detailed hypothesis driven analysis. The last three sub-sections of results (transcriptomic, metabolomic, and phenotype assessment) should have been therefore presented first. Rest of the sub-sections relate to details and can follow the above three.

Reply:

We appreciate this comment. The reviewer recommended to first present three sub-sections of results (transcriptomic, metabolomic, and phenotype assessment). However, it is required to first explain how the condition of restraint stress was decided, so that we first presented this in Fig. 1, then three sub-sections of results (transcriptomic, metabolomic, and phenotype assessment), and others.

Minor comments:

We deeply apologize for poor editing at many places.

1. Lines 1-2: “Paternal restraint stress affects metabolism in the offspring via *Drosophila* ATF-2 in the germ cells” – may be changed to “Paternal restraint stress affects offspring metabolism via ATF-2 dependent mechanisms in the germ cells in *Drosophila*”

Reply:

The title should be 15 words according to the guideline. So, we have changed the title to “Paternal restraint stress affects offspring metabolism via ATF-2 dependent mechanisms in the *Drosophila* germ cells”.

2. Line 65: space before comma!

Reply:

We have corrected this.

3. Line 70: space before comma!

Reply:

We have corrected this.

4. Lines 93-94: “Injection of TNF- α into F0 male mice, known to be..” should be reframed as “In F0 male mice, injection of TNF- α , known to be..”

Reply:

We have corrected this.

5. Line 95: ref 23 should be cited

Reply:

We have corrected this.

6. Line 132: “resuspend 1ml”, “in” is missing

Reply:

We have corrected this.

7. Line 155: “and” is missing before anti-histone

Reply:

We have corrected this.

8. Lines 259-261: it can simply be stated that an in-house developed software was used... instead of referring to a specific software that is unpublished

Reply:

We have changed the sentence to “We determined the death timing of flies using an in-house developed software.”

9. Lines 265-304: Figure 1A is followed by Figure 1C in the text, with Figure B missing?

Reply:

We deeply apologize for this mistake. We have corrected this.

10. Lines 265-304: Text does not clearly indicate the tissue of interest (testicular germ cells). Please mention as appropriate.

Reply:

We have added following description at lines 292.

“We thought that the pRS induced-w expression in F1 offsprings might cause the epigenetic change in testicular germ cells of father fly, and the epigenome status might be transmitted to F1 offsprings.”

11. Lines 289-290: “1st brood but also in the 2nd and 4th broods” – as 3rd brood shows insignificant but higher expression, it may be indicated in the text

Reply:

We appreciate this comment. We have added the description in the text.

12. Lines 295-296: “an rDNA coding region which located in a heterochromatic region” – ‘is’ missing after ‘which’

Reply:

We have corrected this.

13. Line 301: “can induces” – can induce

Reply:

We have corrected this.

14. Lines 311-315: “We observed the F1 offspring from the dATF-2 mutant exhibited high level of w gene expression as described previously (19), and pRS did not further affect the level of w gene expression (Figure 1G,..... Figure 1H”): But the figure legend shows eye pigment levels and not w gene expression for G and H! Also, it is unclear what “control” means in these figure panels. Where has it been shown that dATF-2 exhibit higher level of phenotypic trait?

Reference to previous work, that too in relation to a different mutant type, further compounds the problem. The authors need to clarify all these issues.

Reply:

We apologize for the confusion. We have unified the description to “the eye pigment level”. We have also described the definition of “control” in the legend to Fig. 1a and 1c as follows.

“Levels of red-eye pigment were measured in pRS-exposed and pRS-free (control) w^{m4} F1 progeny.” (Fig. 1a)

“The value of red-eye pigment represents relative to control of F1 male progeny, which derived from RS-free fathers.” (Fig. 1b)

In Supplementary Fig. 2f, we have also added the data indicating that the $dATF-2^{KO}$ mutant showed the increased eye pigment level like $dATF-2^{PB}$ in the background of w^{m4} .

15. Lines 348-350: “we have checked the expression level of these mRNAs by qRT-PCR using three biological replication samples (Figure 3B). We observed both *upd3* and *spz3* made peaks” – language correction required, e.g., we checked.....the expression made peaks is odd

Reply:

We deeply apologize for this mistake. We have corrected sentences as following.

“we checked the expression level of these mRNAs by qRT-PCR using three biological replication samples (Figure 3B). The expression of *upd3* and *spz3* made peaks in the early phase of RS (within 1.5 hours), and then only *upd3* increased gradually during RS.”

16. Line 384: “RS information which involved in dATF-2 pathway-dependent epigenetic change”, please correct the language

Reply:

We have corrected this as follows.

“RS information which causes the dATF-2 pathway-dependent epigenetic change.”

17. Line 387: “because we wanted to analyze under normal chromatin status” – please correct the language

Reply:

We have corrected this as follows.

“To examine this under normal heterochromatin status,”

18. Line 431: “which is involved glycolysis” – language correction required

Reply:

We have corrected this as follows.

“which is involved in glycolysis”

19. Line 510: “repeated exposure to RS three times” – repeated as well as three times!

Language correction needed

Reply:

We have removed “repeated”.

20. Line 540: whichis – space required

Reply:

We have corrected this.

21. Line 585: “Supplementary Data are available at NAR online” – an example of poor editing

Reply:

We deeply apologize for this mistake. We have corrected this.

22. Supplementary Figure 1 legend: the last sentence is confusing. Please state clearly that the same refers to previous work.

Reply:

We apologize for the confusion. We have corrected as follows:

“Note that F1 offsprings from *dATF-2^{KO}* mutant father exhibit high eye-pigment level like *dATF-2^{PB}* previously reported (19), and the eye-pigment level did not increase further in offsprings from RS exposed fathers.”

23. Line 632: stress instead of dtress

Reply:

We have corrected this.

24. Line 549: “A sperm non-coding RNAs may also” – language correction needed

Reply:

We have corrected this as follows.

“Non-coding RNAs in sperm may also”

25. Lines 852-853: Figure 1 legend for panels F, G, and H are wrongly indicated.

Reply:

We have corrected this.

26. Line 857: Figure 2 legend: sense and anti-sense probes are not matching with labelling shown in the figure (lower and upper images)

Reply:

We deeply apologize for this mistake. We have corrected this in the Figure 2B.

27. Line 871: “No that” - Note that

Reply:

We have corrected this.

28. Figure 2: Color scheme for spermatogonia does not seem to match

Reply:

We deeply apologize for this mistake. We have corrected this.

29. Figure 3: Figure 3A may be accommodate in Supplementary Figure 5 as it is a single sample based preliminary screening.

Reply:

We appreciate this comment. We have accommodated Figure 3 in Supplementary Figure 5.

30. Figure 4D: this panel seem unnecessary, given supplementary figure 6.

Reply:

We appreciate this comment. To show which genes are up- and down-regulated by pRS in glycolysis and TCA cycle, we are keen to this Figure. Based on your suggestion, we moved this Figure to Supplementary Figure 6.

31. Figure 7: it seems unnecessary

Reply:

Thank you for the suggestion. Since reviewer 2 showed a very positive response to Figure 7 as described below, however, we would like to keep it.

“The models are especially helpful! Thank you as well for Figure 7 (model).”

32. The term “father flies”, used throughout the manuscript, is odd and may be replaced by fathers or male parents etc.

Reply:

We appreciate this comment. We have replaced the term “father flies” by fathers.

Response to comments by reviewer #2

Major Comments:

1. “However, the following three successive broods did not exhibit higher w gene expression. Exposure to RS twice also yielded similar results (Figure 1D). Repetitive RS exposure three

times resulted in higher w gene expression not only in the 1st brood but also in the 2nd and 4th broods (Figure 1E).”

o This is an especially amazing finding by the authors and needs to be further analyzed.

One might predict that long term memory of the stress has created the maintained inheritance of additional brood, as shown in other *Drosophila* transgenerational studies. I believe it would be important to test some learning and memory genes (i.e. orb2) to see how similar or different this paradigm is. Ideally, a inducible line would be used to knockdown the gene in the father but allow for wild-type levels in offspring. This experiment would also provide evidence to author speculation of the nervous system communicating to the germline.

Reply:

We appreciate this supportive comment. In the present study, we found that pRS affects the epigenetic status of germline cells in testes and repetitive pRS prolonged the effect of pRS. We also showed that the p38-MEKK1-dATF2 pathway is required for pRS effect in germ cells. Our recent studies have also shown that the p38-MEKK1-dATF2 (or mouse homolog ATF7) pathway has a central role in several paternal stress-dependent effects of offsprings not only in *Drosophila* but also in mice (Seong et al., 2011; Yoshida et al., 2020). In the present study, we have reported that RS-induced Upd-3 activates the phosphorylation of p38 in testes and induces an epigenetic change of the offsprings. As shown in the present study, the p38 activation by RS is transient. Because spermatogenesis progresses continuously and the sensitive stage of germ cells to pRS might be limited, the repetitive pRS might be required for prolonged effect by pRS in germline cells.

Reviewer 2 commented that another mechanism, such as machinery of long-term memory, might contribute to the stable epigenetic inheritance of paternal stress. Reviewer 2 recommended the experiment using some learning and memory genes, such as orb2. It is a very intriguing idea that long-lasting memory formation mediated by *Drosophila* CPEB Orb2 contributes to the repetitive pRS dependent epigenetic inheritance. Some reports show that Orb2 prion-like oligomerization upon neuronal stimulation and expression of spliced coding Orb2 mRNA by some behavioral training occurs required for the long-lasting memory formation (Majumdar et al., 2012; Gill et al., 2017; reviewed in Si and Kandel 2016). Moreover, recent reports indicated that ethanol seeking post-wasp exposure is dependent on long term memory formation which mediated by Orb2, and the behavioral change can be transmitted maternally to next-generation (Bozler et al., 2017 and 2019). In our present study, however, we focused on the mechanism of the pRS-induced epigenetic change mainly in

germ cells. Especially, we have shown that Upd-3 expresses by pRS, activates p38 phosphorylation directly in germline cell, and induces transgenerational epigenetic change through the male germ cells. Furthermore, we showed that Upd-3 induction is not increased by repetitive pRS. These findings suggest that the epigenetic change in the germ cells, at least partly, contributes to the effect of pRS on the offspring. So, we would like to test the experiments recommended by Reviewer 2 in our future study.

(References)

Seong KH, Li D, Shimizu H, Nakamura R, Ishii S. Inheritance of stress-induced, ATF-2-dependent epigenetic change. *Cell*. 2011 Jun 24;145(7):1049-61. doi: 10.1016/j.cell.2011.05.029.

Yoshida K, Maekawa T et al. ATF7-dependent epigenetic change is required for intergenerational effect of paternal low-protein diet. *Mol. Cell*. 2020 (Accepted)

Majumdar A, Cesario WC, White-Grindley E, Jiang H, Ren F, Khan MR, Li L, Choi EM, Kannan K, Guo F, Unruh J, Slaughter B, Si K. Critical role of amyloid-like oligomers of *Drosophila* Orb2 in the persistence of memory. *Cell*. 2012 Feb 3;148(3):515-29. doi: 10.1016/j.cell.2012.01.004. Epub 2012 Jan 26.

Gill J, Park Y, McGinnis JP, Perez-Sanchez C, Blanchette M, Si K. Regulated Intron Removal Integrates Motivational State and Experience. *Cell*. 2017 May 18;169(5):836-848.e15. doi: 10.1016/j.cell.2017.05.006.

Si K, Kandel ER. The Role of Functional Prion-Like Proteins in the Persistence of Memory. *Cold Spring Harb Perspect Biol*. 2016 Apr 1;8(4):a021774. doi: 10.1101/cshperspect.a021774. Review.

Bozler J, Kacsoh BZ, Chen H, Theurkauf WE, Weng Z, Bosco G. A systems level approach to temporal expression dynamics in *Drosophila* reveals clusters of long term memory genes. *PLoS Genet*. 2017 Oct 30;13(10):e1007054. doi: 10.1371/journal.pgen.1007054. eCollection 2017 Oct.

Bozler J, Kacsoh BZ, Bosco G. Transgenerational inheritance of ethanol preference is caused by maternal NPF repression. *Elife*. 2019 Jul 9;8. pii: e45391. doi: 10.7554/eLife.45391.

2. The RU486 feeding protocol is not described in the methods (i.e. how much was fed, how often, how was it administered, etc.). I am also unable to find the solvent only control for Figure 3C, the ideal control (rather than 0 microliter/mL).

Reply:

We deeply apologize for insufficient information. We have added the protocol in the method section. We used standard medium with the solvent (100% ethanol) as the control. So, we also corrected Figure 3C.

3. I am fascinated by the observation that the F1 male and female flies have inherited changes, but the F2 do not. I would very much like to see if the effects are additive or an 'all or nothing' result. For example, placing the F1 males into the sandwich and then testing the F2's. Is the phenotype enhanced or the same? I believe that this is a critical question. Either result will place this manuscript on a unique level as being the only manuscript that I am aware of for examining trans and multi generational effects in *Drosophila* as a function of paternal stress. Additive results suggest that further epigenetic changes take place (perhaps further enhancing metabolites in F2). The same result suggests an 'all or nothing' response, which is a curious but exciting evolutionary finding.

Reply:

We appreciate this supportive comment. We performed RS exposure for two consecutive generations and compared the red-eye pigment level with the No RS F1 control and F1 flies from RS exposed father. We observed RS during two consecutive generations did not show any additive or synergistic change of red-eye pigment level. The result suggested RS could affect only the next generation but not further generations.

We have added the data in Supplementary Fig. 1g and the following description at lines 212 in the main text.

"We explored whether the RS-induced eye-pigment phenotype might carry to the F2 generation. pRS did not affect the eye-pigment level in the F2 offspring (Supplementary Fig. 1f), and there was no additive or synergistic effect of RS exposure performed for two consecutive generations (Supplementary Fig. 1g). These results indicate that the pRS-induced heterochromatin disruption is transmitted to the F1, but not to the F2 offspring."

Minor Comments:

1. Line 102: "standard *Drosophila media*"—please indicate recipe/citation for this

Reply:

We have added the recipe as Supplementary Note 1.

2. Methods: what temperature/humidity/light-cycle is the resistant stress performed at?

Reply:

We appreciate this comment. We have added the description in the text.

3. Line 110: “virgin males”—males are ready to mate upon eclosion, so unless they were removed upon eclosion from vials, it is very difficult to make this claim. Consider “day 0” males as an alternative.

Reply:

We appreciate this comment. We have added the description “separated from females on day 0” in the text.

4. PEV assay: Where male and female flies used in the “30 flies”?

Reply:

We apologize for the insufficient description. We have added the description “Eclosed flies were separated by sex and reared in new vials (30 flies/vial) at 25°C” in the text.

5. Line 127: please change “wash” to “washed”

Reply:

We have corrected this.

6. Line 150: “Ten” needs to be lower case to “ten”

Reply:

Reviewer 2 might point out the word “The”. We deeply apologize for this mistake. We have corrected this.

7. Line 260: “sapphire software”—if possible, please elaborate, even if just a little, on how the software works here.

Reply:

As pointed out by reviewer 1, we would like to change the sentence to “We determined the death timing of flies using an in-house developed software.”

8. Line 269: “RS stress”—please consider including a statement on the physiological relevance of the fly sandwich—i.e. how/when/why might this stressor happen in nature (if at all) then how this stress resembles other stressors.

Reply:

We appreciate this comment. However, RS is very artificial stress not only in flies but also in the mammal model. Especially, it is very difficult to mention the stress resembles other stressors in the fruit fly.

9. Figure 1 F demonstrates changes in H3K9me2. The authors find that the F2 generation is not affected why way of variegation. Consider testing the H3K9me2 state in F2—perhaps there is still a marginal change, but not sufficient to induce changes. If there is a change, this could demonstrate a ‘primed’ state, where stressing the F2 could cause a stronger effect than the F0.

Reply:

We appreciate this comment. It is also an important point, but we think that this should be tested in our future study.

10. Line 387: incomplete sentence/ subject verb disagreement

Reply:

We have corrected this.

11. Discussion: Consider discussing other *Drosophila* transgenerational studies in the discussion and highlighting the unique finding in this manuscript.

o J Hered. 2019 May 7;110(3):300-309. doi: 10.1093/jhered/esz009.

♣ Age of Both Parents Influences Reproduction and Egg Dumping Behavior in *Drosophila melanogaster*.

♣ Mossman JA1, Mabeza RMS1, Blake E1, Mehta N1, Rand DM1.

o Elife 2018

♣ Transgenerational dynamics of rDNA copy number in *Drosophila* male germline stem cells

♣ Kevin L Lu, Jonathan O Nelson, George J Watase, Natalie Warsinger-Pepe, Yukiko M Yamashita University of Michigan, United States

o Bozler J (2019) Maternal Priming of Offspring Immune System in *Drosophila*. G3: Genes, Genomes, Genetics. doi: doi.org/10.1534/g3.119.400852. PMID:

o Bozler J (2019) Transgenerational inheritance of ethanol preference is caused by maternal NPF repression eLife. doi: doi.org/10.7554/eLife.45391. PMID: 31287057

Reply:

Thank you for your advice. All of these reports are very interesting. We have referred some of them in introduction. Because we have focused on the epigenetic inheritance in our manuscript, we have discussed only Bozler's papers in discussion.

REVIEWERS' COMMENTS:

Reviewer #2 (Remarks to the Author):

Seong et. al present their work, entitled, "Paternal restraint stress affects metabolism in the
offspring via *Drosophila* ATF-2 in the germ cells" In this study, the authors investigate the role
of environmental stressors on the paternal germline and its subsequent effect on F1 *Drosophila*
offspring. What is very exciting is that this phenotype was not passed onto the F2, but rather,
halted in the F1 state. Interestingly, the stress causes heterochromatic changes in the paternal
genome leading to gene expression changes in the offspring. The key epigenetic factor shown
to be involved is ATF-2, as shown by mutant analysis. Collectively, this body of work identifies
paternal stress induces metabolic changes in offspring that have not experienced the stress,
demonstrating environmental stressors induce epigenetic changes yielding phenotypic
outcomes. This is a truly remarkable observation, and is important to the emerging field of
germline physiology, epigenetics, and transgenerational inheritance. I believe that this
manuscript will draw broad interest from multiple research fields, including neuro science,
epigenetics, evolution and ecology, in addition to being of interest to the public. The
experiments are logically arranged and elegantly executed. Wonderful work by the authors!

It is my recommendation that this important manuscript be accepted. My comments are minor
and primarily text related, in particular the changing of tenses throughout the manuscript need
to be standardized.

My concerns have been very nicely addressed. Most excitingly, their additional experiment
described in S Fig 1F. Eye-pigment level in the F2 offspring shows no additive or synergistic
effect of RS exposure performed for two consecutive generations. My suggestion would be to
place this in the primary figure series, but I leave that at the author's discretion.

Major Comments

Minor Comments

• Line 44: nematoda to nematodes (to correct for plural)

• Tenses in the introduction (and throughout the text) keep shifting from past tense to present
tense. Please keep consistent.

• Line 67: "on the other hands"—please change to "on the other hand"

• Line 69-70: incomplete sentence and subject-verb disagreement:

o "It remains still unclear the molecular mechanisms of how metabolic..."

• Line 160: Add space between words "wm4females"

• Line 164: "ring. (Fig. 1h," remove period between ring and (.

• Line 269-270: There was no difference in the upd3 mRNA level between whole body samples
of wild-type and dATF-2 homozygous mutant (data not shown)

o Please show data

• Figure 2 C: I am having a great deal of difficulty seeing the white numbers on the yellow
background bars. If possible, I would recommend changing the colors for us with older eyes.

• Figure 5B. Please add a scalebar.

• Figure 5 F. Please consider changing the staining color from green to monochromatic. The
stain itself is hard to see and I think the grayscale would help more clearly demonstrate the
stain/finding.

• Figure S 6 A: Please consider changing the various stains, including P-p38 to monochromatic
colors. The stains are difficult to see with the blue-on-black color pattern. Also, please include
a scale bar for this figure.

Balint Z. Kacsoh | Ph.D.

**Response to the reviewer's comments**

We really appreciate the reviewer comments which are helpful to improve the
quality of the manuscript and we deeply apologize for some mistakes. Our reply
to each comment is shown below.

Reviewer's text will be in blue whereas our part will be in black.

**Response to comments by reviewer**

It is my recommendation that this important manuscript be accepted. My
comments are minor and primarily text related, in particular the changing of
tenses throughout the manuscript need to be standardized.

My concerns have been very nicely addressed. Most excitingly, their additional
experiment described in S Fig 1F. Eye-pigment level in the F2 offspring shows
no additive or synergistic effect of RS exposure performed for two consecutive
generations. My suggestion would be to place this in the primary figure series,
but I leave that at the author's discretion.

Reply:

We appreciate this comment. We have placed the results related to
eye-pigment level in the F2 offspring in the primary figure 1f and 1g as the
reviewer recommended.

*Major Comments Minor Comments*

- Line 44: nematoda to nematodes (to correct for plural)

Reply:

We have corrected this.

- • Tenses in the introduction (and throughout the text) keep shifting from
past tense to present tense. Please keep consistent.

Reply:

We deeply apologize for this mistake. We have corrected this.

- • Line 67: “on the other hands”—please change to “on the other hand”

Reply:

We have corrected this.

- • Line 69-70: incomplete sentence and subject-verb disagreement:

o “It remains still unclear the molecular mechanisms of how metabolic...”

Reply:

We deeply apologize for this mistake. Reply:

We have changed the sentence to “However, it remains still unclear how
metabolic and environmental stress can transmit epigenetically to
offsprings in *Drosophila*.”

- • Line 160: Add space between words “wm4females”

Reply:

We have corrected this.

- • Line 164: “ring. (Fig. 1h,” remove period between ring and (.

Reply:

We have corrected this.

• Line 269-270: There was no difference in the upd3 mRNA level between
whole body samples of wild-type and dATF-2 homozygous mutant (data
not shown)

o Please show data

Reply:

We appreciate this comment. We have added the data at Supplementary
Fig. 7b.

• Figure 2 C: I am having a great deal of difficulty seeing the white numbers
on the yellow background bars. I possible, I would recommend changing
the colors for us with older eyes.

Reply:

We appreciate this comment. We have changed the colors to black.

• Figure 5B. Please add a scalebar.

Reply:

We appreciate this comment. We have added the scalebar.

• Figure 5 F. Please consider changing the staining color from green to
monochromatic. The stain itself is hard to see and I think the grayscale
would help more clearly demonstrate the stain/finding.

Reply:

We appreciate this comment. We have changed the staining colors as the
reviewer recommended.

- • Figure S 6 A: Please consider changing the various stains, including
P-p38 to monochromatic colors. The stains are difficult to see with the
blue-on-black color pattern. Also, please include a scale bar for this
figure.

Reply:

We appreciate this comment. We have changed the staining colors as the
reviewer recommended.